# RelayCaching: Accelerating LLM Collaboration via Decoding KV Cache Reuse

**Yingsheng Geng** [1 2]   **Yuchong Gao** [3]   **Weihong Wu** [4]   **Guyue Liu** [5]   **Jiang Liu** [1 2]

## Abstract

The increasing complexity of AI tasks has shifted the paradigm from monolithic models toward multi-agent large language model (LLM) systems. However, these collaborative architectures introduce a critical bottleneck: redundant prefill computation for shared content generated by previous agents, which significantly increases KV cache memory usage and time-to-first-token (TTFT). While various KV cache methods have been proposed to mitigate prefill redundancy, they either fail to maintain accuracy on agent-generated outputs or exhibit low reuse rates due to rigid constraints. We present RelayCaching, a training-free inference method that directly reuses decoding phase KV caches from previous agents in subsequent prefill phases. Our key insight is that KV caches for identical content are highly consistent across phases, while prefix-induced deviations are sparse and localized within a limited range of layers and token positions. By selectively recomputing KV caches at these positions, RelayCaching preserves model accuracy with minimal overhead, yielding a superior accuracy–efficiency trade-off over existing methods. Experiments on diverse collaborative LLM tasks spanning mathematical reasoning, general knowledge, and code generation demonstrate that RelayCaching achieves over 80% KV cache reuse, reduces TTFT by up to $4.7\times$ compared to the standard pipeline, all with negligible accuracy degradation.

## 1. Introduction

Multi-agent systems (MAS) have emerged as a dominant paradigm for deploying LLMs on complex tasks, orchestrating specialized agents that collaborate through structured communication (Wu et al., 2024; Hong et al., 2023; Li et al., 2023). Recent frameworks scale MAS to increasingly sophisticated workflows, ranging from multi-turn software development pipelines (Qian et al., 2024) to scientific discovery conducted by teams of coordinated agents (Su et al., 2025). This trend amplifies both capability and computational demands. In these pipelines, agents exchange intermediate outputs such as reasoning chains, API or code artifacts, and summarized context, enabling task decomposition and iterative refinement (Wu et al., 2024; Hong et al., 2023). As illustrated in Figure 1, consider a development workflow: an Architect agent designs an API specification, a Developer implements it, and a Reviewer evaluates the result. Each upstream output is embedded into downstream prompts, causing cascading redundant prefill.

However, this output-to-input data flow suffers from *prefix variation*: the same tokens appear with different preceding contexts between decoding and prefill phases. This mismatch introduces severe *cascading context redundancy*, as serving systems must treat previously generated text as entirely new input and recompute KV caches from scratch during prefill. In a pipeline with $M$ interaction turns, each agent must reprocess the accumulated history from all preceding interactions, causing cumulative prefill cost to grow quadratically with interaction turns ($O(M^2)$) (Ye et al., 2026). This redundant prefill computation for shared content becomes a critical bottleneck in multi-agent pipelines, significantly increasing KV-cache memory usage and TTFT.

Existing prefill redundancy optimization methods based on *prefix caching* and *pre-computed caching* are inadequate for the dynamic nature of MAS. *Prefix caching* (Kwon et al., 2023; Zheng et al., 2024) relies on strict positional alignment and fails in MAS, where reused content frequently appears at non-prefix positions. *Pre-computed caching* (Gim et al., 2024; Yao et al., 2025) enables non-prefix reuse but assumes static, offline content (e.g., documents in RAG), limiting applicability in MAS where outputs are generated on the fly. Consequently, current systems often fall back to expensive full prefilling in MAS settings (Figure 1(a,b)).

This limitation motivates a fundamental question: *can we directly reuse decoding KV caches to bypass redundant prefill computation despite prefix variation?* To answer this, we

[1]Beijing University of Posts and Telecommunications [2]State Key Laboratory of Networking and Switching Technology, Beijing, China [3]Tsinghua University [4]University of Electronic Science and Technology of China [5]Peking University. Correspondence to: Weihong Wu <wuweihong@uestc.edu.cn>, Guyue Liu <guyue@pku.edu.cn>.

*Proceedings of the 43rd International Conference on Machine Learning*, Seoul, South Korea. PMLR 306, 2026. Copyright 2026 by the author(s).

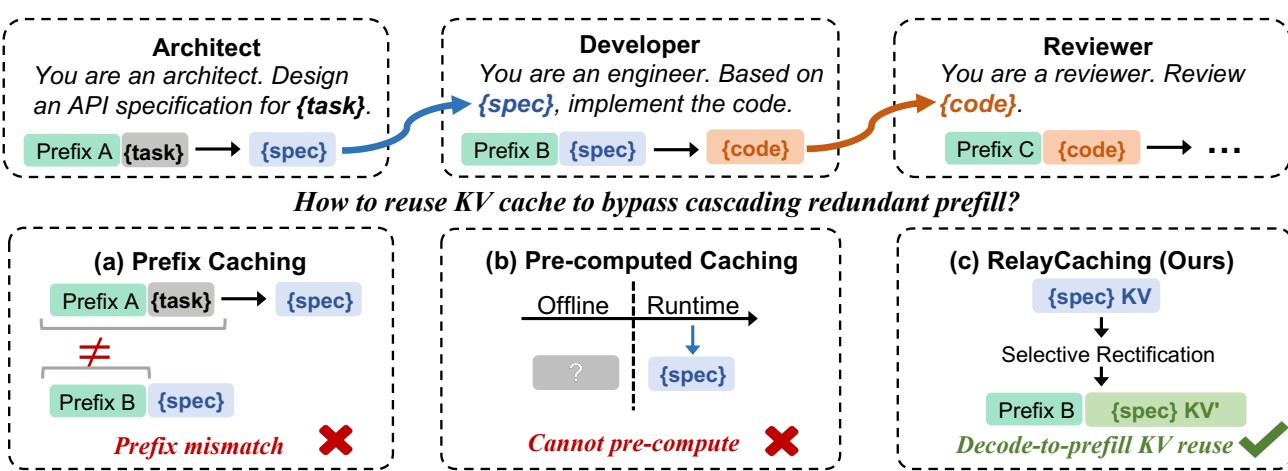

*Figure 1.* Overview of KV cache reuse in multi-agent LLM collaboration. **Top:** A typical workflow where an Architect generates {spec}, a Developer produces {code} based on it, and a Reviewer evaluates the result, so each agent's output becomes the next agent's input, causing cascading redundant prefill. **Bottom:** Comparison of caching strategies. (a) Prefix Caching fails due to prefix mismatch; (b) Pre-computed Caching cannot handle dynamic content; (c) RelayCaching enables decode-to-prefill KV reuse via selective rectification.

conduct a systematic empirical study comparing decoding KV caches with their full-prefill counterparts. Our analysis reveals three key findings that establish the feasibility of direct reuse with targeted rectification: **(1) High macro-level alignment.** Despite prefix variation, decoding KV caches maintain high alignment with full-prefill counterparts, with value cosine similarity emerging as the primary deviation indicator. **(2) U-shaped layer-wise similarity profile.** Middle layers show the largest deviations and dominate subsequent generation quality, while shallow and deep layers remain relatively stable. **(3) Sparse and correlated token-wise deviations.** Token-wise deviations are sparse and exhibit strong inter-layer correlation.

Building on these observations, we propose *RelayCaching* a training-free method that directly reuses decoding KV caches from previous agents in prefill phases. RelayCaching comprises two core components: (1) a *layer-range profiler* that identifies the critical layer range using the U-shaped profile and selects a detection layer based on inter-layer correlation, and (2) a *token selector* that combines deviation-based and influence-based selection to pinpoint tokens requiring rectification. This design enables RelayCaching to maintain accuracy comparable to full prefilling while rectifying only a small fraction of KV caches, yielding a superior accuracy–efficiency trade-off over existing methods.

In summary, this paper makes the following contributions:

- We systematically characterize KV deviations between decoding and prefill phases, revealing that decoding KV caches exhibit high alignment with full-prefill counterparts despite prefix variation. Residual deviations follow systematic patterns that enable targeted rectification: U-shaped similarity across layers and sparse, inter-layer-correlated deviations across tokens.

- We propose RelayCaching, a training-free method that directly reuses decoding KV caches from upstream agents in downstream prefill phases. RelayCaching employs layer-range profiling to confine rectification to middle layers and token selection to identify critical tokens, achieving efficient cache alignment with minimal overhead.

- We demonstrate that RelayCaching maintains generation quality comparable to full prefilling across diverse tasks while achieving over 80% KV cache reuse and up to $4.7\times$ TTFT reduction in multi-agent systems.

## 2. Related Works

### 2.1. LLM-Based Multi-Agent Workflows

LLM-based Multi-Agent Systems have emerged as a prevalent paradigm for solving complex tasks through collective intelligence. Foundational frameworks such as CAMEL (Li et al., 2023) and AutoGen (Wu et al., 2024) pioneered role-playing for task decomposition, while MetaGPT (Hong et al., 2023) introduced Standard Operating Procedures for assembly-line workflows. Recent frameworks like GPTSwarm (Zhuge et al., 2024) and MegaAgent (Wang et al., 2025a) further model agent collectives as dynamic computation graphs. However, these pipelines suffer from *cascading context redundancy*, where upstream outputs are repeatedly reused under changing prefixes, forcing KV cache rebuilding and incurring significant overhead.

### 2.2. Efficient LLM Inference and KV Caching

KV caching has become the de facto standard for mitigating redundant prefill computation. We categorize existing approaches into prefix caching and pre-computed caching. Prefix caching reuses KV caches for recurring, identical

prefixes. Systems like vLLM (Kwon et al., 2023) and SGLang (Zheng et al., 2024) introduce PagedAttention and RadixAttention to share prefix KV caches across requests. Pre-computed caching relaxes the strict prefix constraint by encoding reusable contexts in advance. Prompt Cache (Gim et al., 2024) enables reuse for frequent prompt modules; Block-Attention (Ma et al., 2025) reuses KV caches for static documents with fine-tuning. KVLink (Yang et al., 2026) and APE (Yang et al., 2025b) align independently encoded KV caches via trainable tokens or adaptive scaling; CacheBlend (Yao et al., 2025) and EPIC (Hu et al., 2025) correct precomputed caches through dynamic or static token recomputation; DroidSpeak (Liu et al., 2024) extends pre-computed caching across models with shared architectures. Despite these advances, prefix caching requires strict positional alignment that agent-generated contexts rarely satisfy, while pre-computed caching assumes static, offline-encodable content, limiting its applicability to dynamically generated outputs in multi-agent settings. These limitations motivate *decode-to-prefill KV reuse*, which transfers KV caches from one agent's decoding phase to another's prefilling phase. This paradigm enables targeted rectification of runtime-generated KV caches without prefix alignment constraints, preserving accuracy with minimal overhead.

## 2.3. Efficiency in Collaborative Inference

Recent efforts to accelerate collaborative inference focus on two orthogonal aspects: topology sparsification and state reuse. Topology sparsification aims to reduce the volume of agent interactions. AgentPrune (Zhang et al., 2025) and AgentDropout (Wang et al., 2025b) optimize message-passing graphs by pruning redundant agents, but treat model inference as a black box. State reuse seeks to minimize computational cost per interaction. To the best of our knowledge, KVCOMM (Ye et al., 2026) is the first to explicitly address *cascading context redundancy* via retrieval-based anchor pools, but relies on external samples and suffers from degraded reuse rates at scale. RelayCaching instead directly exploits the inherent similarity between decoding and prefill KV caches, enabling fine-grained rectification at layer and token granularity without external reference.

## 3. Observations

In this section, we investigate the alignment and deviation between prefill and decoding KV caches under *decode-to-prefill KV reuse*, where the same tokens appear under different preceding contexts. We address two questions: (1) *Reusability*: are decoding KV caches sufficiently aligned with full-prefill counterparts despite prefix variation? (2) *Deviation patterns*: do residual deviations exhibit systematic patterns that can guide efficient rectification? We conduct experiments on 2WikiMQA (Ho et al., 2020) with Mistral-

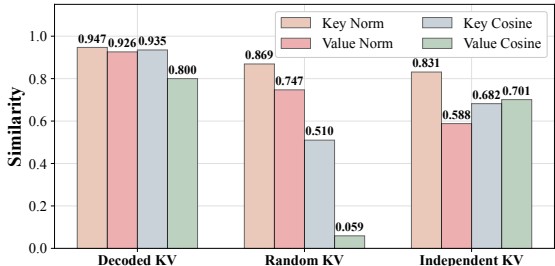

*Figure 2.* **Macro-level KV similarity.** Average cosine and norm similarity of *Decoding KV*, *Random KV*, and *Independent KV* against *Full-Prefill KV*, averaged across all layers and tokens on 2WikiMQA.

7B-Instruct-v0.3 (Jiang et al., 2023) using a summarize-then-answer workflow. We further verify that our findings generalize to other model architectures in the Appendix A.

## 3.1. Macro-Level KV Similarity

We first validate that decoding KV caches remain globally aligned with full-prefill counterparts, finding that *value cosine similarity* emerges as the primary deviation signal. We compare cosine and norm similarity under three settings: (1) *Decoding KV*, produced during decoding with shifted prefixes; (2) *Random KV*, from prefill with the summary replaced by random tokens; and (3) *Independent KV*, from prefill on summary tokens alone without preceding prefix. As shown in Figure 2, *Decoding KV* maintains high macro-level similarity, whereas *Random KV* shows much lower similarity with value cosine approaching zero. Notably, keys in *Decoding KV* align closely with full-prefill keys in both direction and magnitude, whereas values show directional perturbations while largely preserving their magnitudes. In contrast, *Independent KV* exhibits a distinct deviation pattern, indicating that pre-computation schemes relying on independently encoded KV caches are incompatible with *decode-to-prefill KV reuse*.

**Insights.** Decoding KV caches remain highly aligned with full-prefill counterparts, validating the reuse premise. Value cosine similarity is the lowest among all metrics, so we focus on it in subsequent layer- and token-wise analysis.

## 3.2. Layer-Wise Deviation Pattern

We next analyze how deviations distribute across layers, finding that middle layers exhibit the largest deviations and contribute most to subsequent generation errors. Figure 3a reveals a characteristic *U-shaped profile* in value cosine similarity: high in shallow layers, dropping to a minimum in middle layers, and partially recovering in deeper layers. To verify that middle-layer deviations are the primary error source for subsequent generation, we perform an oracle experiment that selectively substitutes decoding KV caches with full-prefill KV at different layer ranges. Fig-

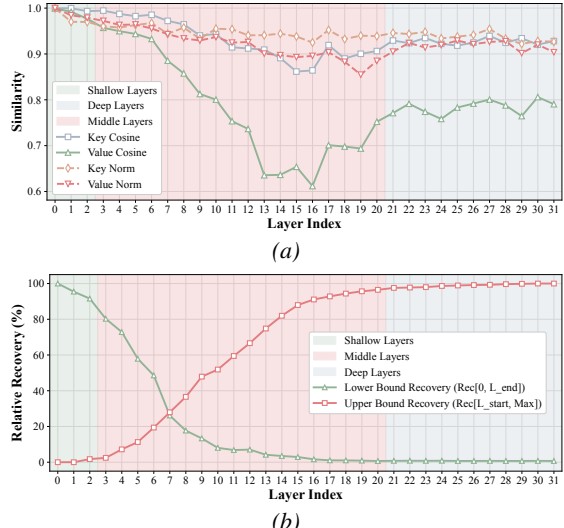

*(a)*

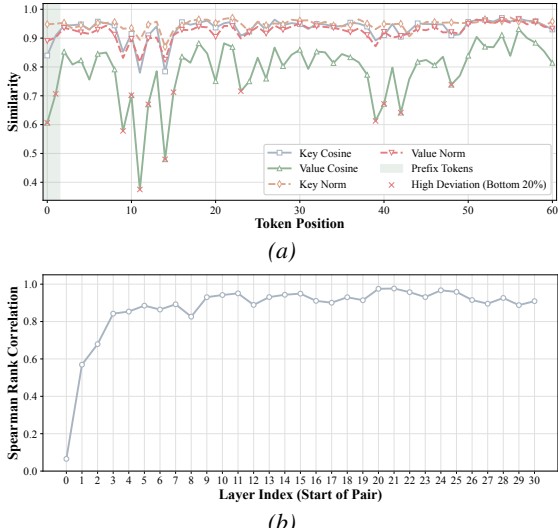

*(a)*

*(b)*

*Figure 3.* **Layer-wise structured deviations.** *(a)* Layer-wise similarity between decoding and full-prefill KV caches, highlighting a U-shaped value cosine similarity profile across layers. *(b)* Relative recovery of value cosine similarity for tokens after the reuse position when decoding KV caches are replaced with their full-prefill counterparts either up to or from a given layer index.

ure 3b reports the resulting value cosine similarity for tokens following the reused segment under two settings: (i) substituting all layers up to index $i$, and (ii) substituting all layers from $i$ onward. In both cases, similarity recovery is steepest when the substituted range covers the middle layers identified by the U-shaped profile, with diminishing returns when extending into shallow or deep layers, confirming that middle layers dominate downstream generation quality.

**Insights.** The U-shaped profile provides a principled criterion for identifying the critical layer range, enabling targeted rectification that captures most of the benefit while bypassing shallow and deep layers.

### 3.3. Token-Wise Deviation Pattern

Finally, we investigate how deviations distribute across tokens, finding that token-level deviations are sparse and exhibit strong inter-layer correlation. As shown in Figure 4a, only a small subset of positions exhibits large deviations, while most tokens maintain high similarity with full-prefill counterparts. Some of these high-deviation tokens are position-dependent, such as tokens near the start of the reuse segment, which are inherently more sensitive to prefix variation; others are content-dependent, reflecting input-specific sensitivity to the preceding context. Figure 4b shows that the Spearman rank correlation of token-wise deviations between adjacent layers starts low but rapidly increases and plateaus, indicating that high-deviation positions persist once they emerge.

**Insights.** The combination of token-wise sparsity and inter-layer correlation implies that rectification can be highly

*(b)*

*Figure 4.* **Token-wise structured deviations.** *(a)* Token-wise similarity between decoding and full-prefill KV caches, averaged over layers, revealing a sparse set of high-deviation positions. *(b)* Spearman rank correlation of token-wise value cosine deviations between adjacent layers, showing that high-deviation positions, once emerged, tend to persist.

selective: by identifying high-deviation tokens at a layer where rankings have stabilized, we can propagate this selection to subsequent layers, achieving most of the rectification benefit at minimal cost.

## 4. RelayCaching

### 4.1. Overall Architecture

Guided by Section 3, we present RelayCaching, a training-free method that accelerates multi-agent collaboration by reusing decoding KV caches with targeted rectification. The key insight is that deviations concentrate in middle layers and a sparse set of tokens, enabling accurate repair at these positions instead of full-prefill recomputation. As depicted in Figure 5, RelayCaching comprises two components: a layer-range profiler that exploits the U-shaped similarity profile and inter-layer correlation identifies the critical layer range $[L_{\text{start}}, L_{\text{end}}]$ and a detection layer $L_{\text{det}}$; and a token selector that combines deviation-based and influence-based criteria. Together, these enable selective recomputation of identified tokens within the critical range, preserving generation quality at a fraction of the cost.

### 4.2. Layer-Range Profiler

The layer-range profiler determines *where* to apply rectification and *at which layer* to perform token selection. The U-shaped profile (Section 3.2) suggests rectification should concentrate on middle layers $[L_{\text{start}}, L_{\text{end}}]$, while inter-layer correlation (Section 3.3) enables identifying a detection layer $L_{\text{det}}$ for reliable token selection. The profiler performs

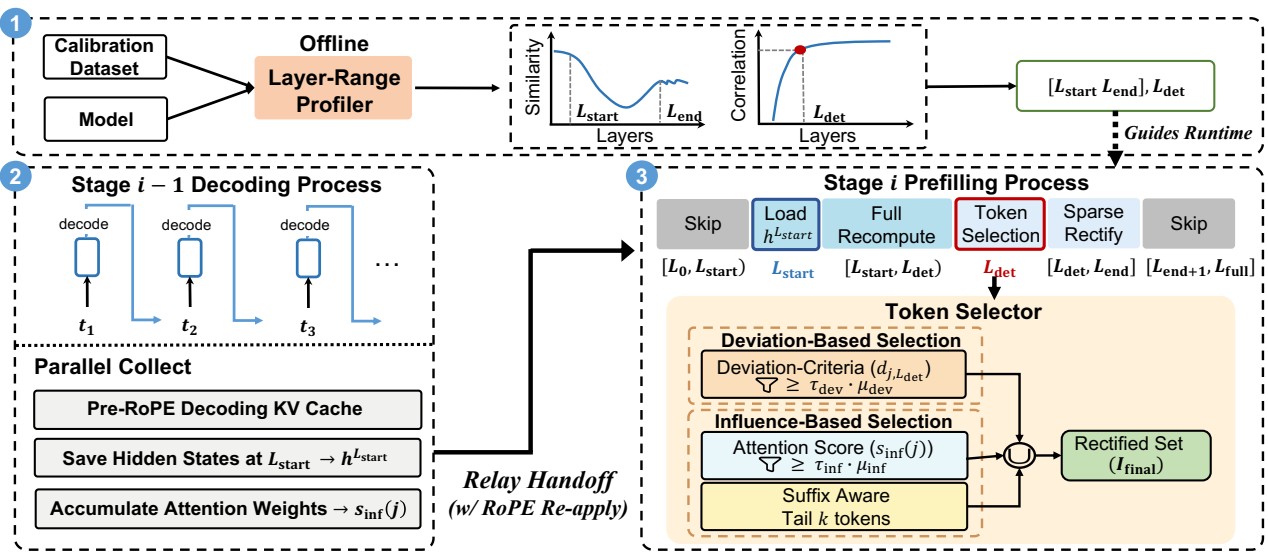

*Figure 5.* **Overview of RelayCaching.** (1) An offline layer-range profiler analyzes calibration data to identify a critical layer range $[L_{\text{start}}, L_{\text{end}}]$ from the similarity profile and a detection layer $L_{\text{det}}$ from the inter-layer correlation profile. (2) During stage-$(i-1)$ decoding, the cache manager stores pre-RoPE KV caches, saves hidden states at $L_{\text{start}}$, and accumulates influence scores $s_{\text{inf}}(j)$. (3) At stage-$i$ prefilling, RelayCaching re-applies RoPE to align reused KV caches to new positions, performs token selection at $L_{\text{det}}$ via combined deviation- and influence-based criteria, and finally applies sparse rectification on selected tokens within $[L_{\text{start}}, L_{\text{end}}]$.

offline profiling on auxiliary datasets to determine these parameters. Let $\mathbf{I}_{\text{reuse}}$ denote the set of token positions in the reuse segment. To quantify the alignment between reused and full-prefill KV caches, we define the token-wise deviation at position $j$ and layer $\ell$ as

$$d_{j,\ell} = 1 - \frac{1}{H} \sum_{h=1}^{H} \cos(\mathbf{v}_{h,j,\ell}^{\text{reuse}}, \mathbf{v}_{h,j,\ell}^{\text{full}}), \quad (1)$$

where $H$ is the number of heads, and $\mathbf{v}_{h,j,\ell}^{\text{reuse}}, \mathbf{v}_{h,j,\ell}^{\text{full}}$ denote the value vectors of head $h$ at position $j$ in layer $\ell$ under reused decoding and full-prefill, respectively. The layer-wise similarity is then the average over all reused tokens:

$$s_\ell = \frac{1}{|\mathbf{I}_{\text{reuse}}|} \sum_{j \in \mathbf{I}_{\text{reuse}}} (1 - d_{j,\ell}). \quad (2)$$

**Start layer.** Since shallow layers already exhibit high similarity and require no rectification, they can provide a high-fidelity starting point for subsequent computation. We set $L_{\text{start}}$ to the deepest layer with $s_\ell \geq \tau_{\text{st}}$. At inference time, we load cached hidden states $\mathbf{h}^{(L_{\text{start}})}$ as the starting point for rectification rather than recomputing shallow layers.

**End layer.** Deeper layers enter a stable regime where deviations partially recover and additional rectification yields diminishing returns on output quality. We locate the onset of this regime by scanning forward from the minimum-similarity layer $\ell_{\text{min}} = \arg\min_\ell s_\ell$. We estimate a baseline $b = \mu_{\text{last}} - \sigma_{\text{last}}$, where $\mu_{\text{last}}$ and $\sigma_{\text{last}}$ are the mean and standard deviation of $s_\ell$ over the last $T$ layers. $L_{\text{end}}$ is set to the

first layer followed by $C$ consecutive layers satisfying

$$s_\ell \geq b \quad \text{and} \quad |s_\ell - s_{\ell-1}| < \lambda \sigma_{\text{last}}. \quad (3)$$

**Detection layer.** While the layer range defines *where* to rectify, we also need to determine *at which layer* to select tokens to rectify. Detecting tokens at $L_{\text{start}}$ may lack discriminative power due to high similarity in shallow layers. To address this, we identify a detection layer, $L_{\text{det}}$ within $[L_{\text{start}}, L_{\text{end}}]$ where deviation patterns have stabilized. We examine the Spearman rank correlation $\rho_\ell$ of token-wise deviations between adjacent layers. We define the second-order difference of the correlation trend as:

$$\alpha_\ell = \rho_\ell - 2\rho_{\ell-1} + \rho_{\ell-2} \quad (4)$$

We then identify $\ell^*$ as the first layer where $\alpha_\ell$ transitions from positive to negative, indicating that correlation growth has begun to decelerate. The detection layer is set to $L_{\text{det}} = \ell^* + 1$, where the deviation ranking has fully stabilized.

At inference time, layers outside $[L_{\text{start}}, L_{\text{end}}]$ directly reuse decoding KV caches without rectification. Within this range, layers in $[L_{\text{start}}, L_{\text{det}})$ are fully recomputed starting from cached hidden states at $L_{\text{start}}$ to enable deviation detection, while layers in $[L_{\text{det}}, L_{\text{end}}]$ apply sparse token-selective rectification only on selected tokens.

### 4.3. Token Selection Module

Section 3.3 reveals that deviations are *sparse* and *inter-layer correlated*, enabling single-pass token identification

at $L_{\text{det}}$ that propagates throughout $[L_{\text{det}}, L_{\text{end}}]$. Beyond deviation magnitude, a token's importance also depends on its *downstream influence*: moderately deviated tokens can substantially impact outputs if subsequent tokens repeatedly attend to them. Accordingly, RelayCaching selects tokens using two criteria: Deviation-based selection targets tokens with large $d_{j,L_{\text{det}}}$; Influence-based selection prioritizes tokens receiving high attention from subsequent positions. Their union forms the final rectification set.

### 4.3.1. DEVIATION-BASED SELECTION

Deviation-based selection targets tokens with large directional mismatches at $L_{\text{det}}$. Using $s_{\text{dev}}(j) = d_{j,L_{\text{det}}}$ and mean $\mu_{\text{dev}}$, we select:

$$\mathbf{I}_{\text{dev}} = \{j \in \mathbf{I}_{\text{reuse}} \mid s_{\text{dev}}(j) \geq \tau_{\text{dev}} \cdot \mu_{\text{dev}}\}, \quad (5)$$

where $\tau_{\text{dev}}$ is a scaling factor. This mean-relative threshold adapts to sample-specific deviation scales and selects high-deviation positions without imposing a fixed token budget or relying on sorting-based selection. Due to inter-layer correlation, this selection is reused throughout $[L_{\text{det}}, L_{\text{end}}]$.

### 4.3.2. INFLUENCE-BASED SELECTION

Deviation-based selection may miss tokens with moderate deviation but substantial downstream influence. We incorporate influence-based selection to identify such tokens.

**Attention-derived importance.** We measure influence by accumulating attention weights received across all layers and decoding steps:

$$s_{\text{inf}}(j) = \sum_{t,\ell,h} \alpha_{t,\ell,h,j}, \quad (6)$$

where $\alpha_{t,\ell,h,j}$ is the attention weight at decoding step $t$, layer $\ell$, head $h$. We select influential tokens via

$$\mathbf{I}_{\text{inf-score}} = \{j \in \mathbf{I}_{\text{reuse}} \mid s_{\text{inf}}(j) \geq \tau_{\text{inf}} \cdot \mu_{\text{inf}}\}, \quad (7)$$

where $\mu_{\text{inf}}$ is the mean influence score and $\tau_{\text{inf}}$ is a scaling factor.

**Suffix-aware correction.** Attention-derived importance is inherently biased toward early positions. Meanwhile, suffix tokens near the reuse segment boundary often exert disproportionate influence on newly appended content due to attention locality. To balance this, we additionally include the last $K_{\text{suf}}$ trailing tokens as $\mathbf{I}_{\text{inf-suffix}}$, yielding:

$$\mathbf{I}_{\text{inf}} = \mathbf{I}_{\text{inf-score}} \cup \mathbf{I}_{\text{inf-suffix}}. \quad (8)$$

The final set combines both criteria:

$$\mathbf{I}_{\text{final}} = \mathbf{I}_{\text{dev}} \cup \mathbf{I}_{\text{inf}}. \quad (9)$$

This dual-criterion selection maintains a small rectification set while effectively correcting deviations, achieving a favorable accuracy–efficiency trade-off.

## 5. Evaluation

This section presents analyses to answer the following research questions: **RQ1**: Can RelayCaching maintain generation quality comparable to full prefilling? **RQ2**: Can RelayCaching improve efficiency for real-world multi-agent tasks? **RQ3**: How does each component contribute to the efficiency-accuracy trade-off? **RQ4**: How sensitive is RelayCaching to key hyperparameters? **RQ5**: How large is the runtime memory and communication overhead associated with the auxiliary states required for rectification?

### 5.1. Experiment Setup

**Datasets and Workflows.** Following KVCOMM (Ye et al., 2026), we evaluate fully connected multi-agent workflows on GSM8K (Cobbe et al., 2021), HumanEval (Chen, 2021), and MMLU (Hendrycks et al., 2021). We use AIME24 (Zhang & Math-AI, 2024) for controlled efficiency evaluation. Benchmark details and agent configurations are provided in Appendix B.4.

**Models and Implementation.** We use Llama-3.1-8B-Instruct (Dubey et al., 2024) for reasoning and knowledge tasks, Qwen2.5-Coder-7B-Instruct (Hui et al., 2024) for coding, and Qwen3-0.6B (Yang et al., 2025a) for controlled efficiency experiments. All models run in BF16 on NVIDIA H20 GPUs. Unless otherwise specified, we use greedy decoding; stochastic decoding settings are provided in Appendix B.4.2. The layer-range profiler is calibrated on a 10% random split of 2WikiMQA (Ho et al., 2020) for task-agnostic deviation detection. We use default hyperparameters $\tau_{\text{inf}}$=1.45, $\tau_{\text{dev}}$=1.5, and $L_{\text{suf}}$=10.

**Baselines.** We compare against two reference baselines: *Full prefill* (Full), which performs no KV reuse, and *Direct Reuse* (Zero), which reuses upstream KV caches after RoPE correction but without further rectification. We further include three KV-reuse methods: CacheBlend (Yao et al., 2025) and EPIC (Hu et al., 2025) for pre-computed KV caching, and KVCOMM (Ye et al., 2026) for multi-agent reuse. Baseline details are provided in Appendix B.3.

**Metrics.** We report task-specific quality metrics, including pass@1 for HumanEval and accuracy for GSM8K and MMLU. For efficiency, we report *Reuse Rate*, the fraction of layer-token KV entries reused without recomputation, as a length-robust proxy when agent outputs vary across methods. TTFT is reported in controlled efficiency experiments where context lengths are aligned.

### 5.2. Main Results

Table 1 and Figure 6 summarize the quality and reuse-efficiency results across GSM8K, MMLU, and HumanEval. Overall, RelayCaching achieves quality comparable to full

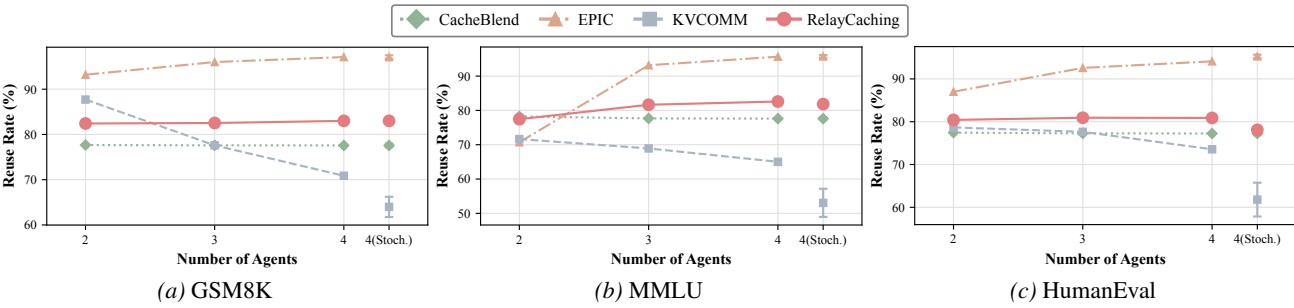

*Figure 6.* **Computational reuse efficiency across diverse benchmarks.** We report the Reuse Rate on *(a)* GSM8K, *(b)* MMLU, and *(c)* HumanEval for 2, 3, and 4 agents under greedy decoding, and 4 agents under stochastic decoding (mean±std over three seeds).

*Table 1.* **Performance comparison on GSM8K, MMLU, and HumanEval.** Results are reported for 2, 3, and 4 agents under greedy decoding, and 4 agents under stochastic decoding (mean±std over three seeds). Among non-reference methods, **bold** and underline mark the best and second-best results.

| Dataset | Method | 2 Agents | 3 Agents | 4 Agents | 4 Agents Stoch. |
|---|---|---|---|---|---|
| *Llama-3.1-8B-Instruct (Accuracy %)* | | | | | |
| GSM8K | Full | 85.97 | 84.69 | 84.69 | 84.26±0.31 |
| | Zero | 40.33 | 40.03 | 53.68 | 45.94±1.51 |
| | CacheBlend | 76.72 | 77.56 | 74.73 | 72.81±1.01 |
| | EPIC | 58.73 | 66.46 | 58.42 | 55.68±0.70 |
| | KVCOMM | 84.23 | 82.34 | 84.91 | 82.89±0.56 |
| | RelayCaching | **84.84** | **85.50** | **85.17** | **85.04±0.31** |
| MMLU | Full | 71.24 | 73.20 | 66.67 | 68.95±0.46 |
| | Zero | 67.97 | 41.18 | 35.29 | 33.55±2.00 |
| | CacheBlend | **69.28** | 64.71 | 66.67 | 55.34±0.34 |
| | EPIC | **69.28** | 45.10 | 43.79 | 39.00±1.00 |
| | KVCOMM | 68.63 | 69.28 | 66.01 | **64.71±2.36** |
| | RelayCaching | **69.28** | **71.90** | **67.97** | 64.71±0.66 |
| *Qwen2.5-Coder-7B-Instruct (Pass@1 %)* | | | | | |
| HumanEval | Full | 82.61 | 85.71 | 87.58 | 82.78±0.69 |
| | Zero | 81.99 | 85.09 | 86.96 | 80.30±0.44 |
| | CacheBlend | **82.61** | 84.47 | 85.71 | 81.49±0.88 |
| | EPIC | 81.99 | 83.23 | 86.34 | 80.43±1.20 |
| | KVCOMM | **82.61** | 83.23 | 85.09 | 78.57±0.44 |
| | RelayCaching | **82.61** | **85.09** | **87.58** | **82.82±0.29** |

prefill while maintaining $\geq 80\%$ KV reuse across most settings. Notably, the layer range detected on 2WikiMQA generalizes across all benchmarks without per-task tuning. Precomputed caching methods EPIC and CacheBlend struggle with the dynamic prefix variations in decode-to-prefill reuse under greedy decoding. EPIC relies on static prefix recomputation and cannot adapt to content-sensitive deviations, with accuracy ranging from 58.42% to 66.46% on GSM8K, far below full prefill. CacheBlend uses value-norm-based detection, which fails to capture the value-direction mismatch that governs decode-to-prefill KV deviations, resulting in accuracy at most 77.56% on GSM8K and 69.28% on MMLU. In the HumanEval setting, the small Zero–Full gap indicates that direct reuse is less harmful. Nevertheless, the results evaluate correction quality: under greedy decoding, prior KV-reuse baselines can fall below Zero when their correction strategies are misaligned with decode-to-prefill deviations, whereas RelayCaching remains comparable to full

*Table 2.* **Per-agent TTFT breakdown and speedup.** Prefix: 512 tokens; Output: 2,048 tokens; Model: Qwen3-0.6B.

| Method | Agent 2 | Agent 3 | Agent 4 | Agent 5 |
|---|---|---|---|---|
| Full | 85.2 | 180.2 | 321.5 | 493.9 |
| CacheBlend | 44.9 | 76.7 | 112.5 | 159.3 |
| Speedup | 1.90× | 2.35× | 2.86× | 3.10× |
| RelayCaching | 40.6 | 57.0 | 73.2 | 104.8 |
|   Full recompute | 1.6 | 2.8 | 4.3 | 7.9 |
|   Index selection | 0.5 | 0.6 | 0.7 | 0.8 |
| Speedup | **2.10×** | **3.16×** | **4.39×** | **4.71×** |

prefill. KVCOMM achieves competitive accuracy across decoding settings, but its similarity-based retrieval leads to less stable reuse. On GSM8K, its reuse rate drops from 88% to 70% as the number of agents increases under greedy decoding. This sensitivity becomes more pronounced under stochastic decoding, where KVCOMM's reuse rate falls to 53–64% across benchmarks, whereas RelayCaching preserves 78–83% reuse and remains stable across both decoding regimes. In summary, RelayCaching achieves a favorable accuracy–efficiency trade-off, preserving quality comparable to full prefill while maintaining high KV reuse across decoding settings.

### 5.3. Efficiency Evaluation

We conduct a controlled efficiency study on AIME using Qwen3-0.6B, where long reasoning traces enable us to stress-test decode-to-prefill KV reuse under growing contexts. We focus on TTFT reduction and its scalability as the number of agents and the reused sequence length increase.

**Latency Breakdown Analysis.** Table 2 decomposes the TTFT latency. As context accumulates from Agent 2 to Agent 5, full prefill latency increases from 85.2 ms to 493.9 ms (5.8×), reflecting the quadratic complexity of prefill computation. In contrast, RelayCaching's latency grows from 40.6 ms to only 104.8 ms (2.6×), achieving a 4.71× speedup at Agent 5. The breakdown reveals favorable scaling characteristics. Index selection grows minimally from 0.5 to 0.8 ms across agents, validating the efficiency of our token selector. The full-recompute cost (1.6–7.9 ms) represents the primary overhead of our method yet remains

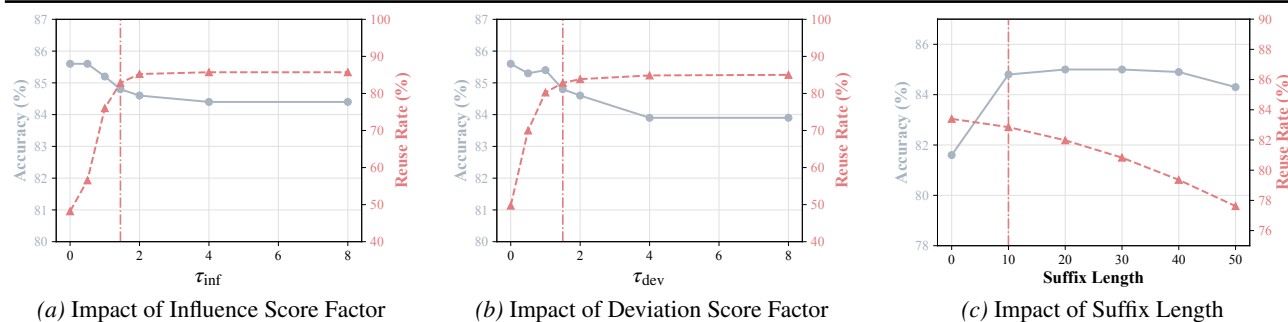

*(a)* Impact of Influence Score Factor     *(b)* Impact of Deviation Score Factor     *(c)* Impact of Suffix Length

*Figure 7.* **Sensitivity analysis on key RelayCaching hyperparameters.** We analyze the trade-off between Accuracy (left axis) and Reuse Rate (right axis) by varying: *(a)* the influence score factor $\tau_{inf}$, *(b)* the deviation score factor $\tau_{dev}$, and *(c)* the suffix length $L_{suf}$. Red markers denote the default configuration used in the main experiments.

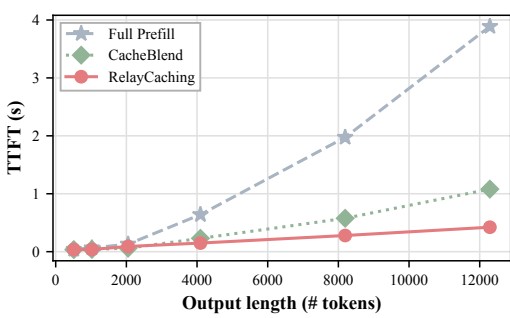

*Figure 8.* **Scalability of TTFT with context growth.** We measure the average TTFT of downstream agents as the cumulative output length scales from 512 to 12,288 tokens.

negligible compared to full prefill. Overall, RelayCaching's latency scales at $2.6\times$, compared to $5.8\times$ for full prefill and $3.5\times$ for CacheBlend, demonstrating sub-linear scaling as the number of agents increases.

**Scalability with Context Length.** To assess performance in long-context regimes, we sweep the cumulative output length from 512 to 12,288 tokens among three collaborating agents. Figure 8 reports the average per-agent TTFT. Standard full prefill exhibits quadratic complexity, with latency escalating to 3.89 s at 12k tokens. While CacheBlend mitigates this growth, it still reaches 1.08 s because it recomputes a fixed proportion of tokens, causing its cost to scale linearly with context length. In contrast, RelayCaching exhibits sub-linear scaling, recording only 0.42 s at the maximum length, achieving a $9.2\times$ speedup over full prefill and $2.5\times$ over CacheBlend. This sub-linear scalability stems from the sparsity of token-wise deviations: as context grows, the number of high-deviation tokens increases more slowly than the total sequence length, keeping the rectification cost sub-linear.

## 5.4. Ablation Results

We conduct a fine-grained ablation on GSM8K under a 2-agent setting (Table 3). Restricting rectification to the critical layer range yields 85.90% accuracy but a modest 46.88% reuse rate. Adding deviation-based selection $\mathbf{I}_{dev}$ boosts reuse to 90.37% but degrades accuracy to 79.68%, indicating that deviation metrics alone overlook generation-

*Table 3.* **Ablation study on GSM8K (Llama-3.1-8B-Instruct).**

| Method | Reuse (%) | Acc (%) |
|---|---|---|
| No Reuse(Full prefill) | 0.00 | 85.97 |
| + Layer Range Profiler | 46.88 | 85.90 |
| + $\mathbf{I}_{dev}$ | 90.37 | 79.68 |
| + $\mathbf{I}_{inf\text{-}score}$ | 85.84 | 81.65 |
| + $\mathbf{I}_{inf\text{-}suffix}$ **(Ours)** | **85.35** | **84.84** |

critical tokens. Further incorporating score-based influence selection $\mathbf{I}_{inf\text{-}score}$ improves accuracy to 81.65% at the cost of lower reuse (85.84%), confirming that influence-aware criteria capture tokens missed by deviation alone. The full RelayCaching, integrating all three strategies, achieves 84.84% accuracy while maintaining 85.35% reuse. This shows that multi-dimensional selection is essential for preserving generation quality.

## 5.5. Sensitivity Analysis

We study the impact of the main RelayCaching hyperparameters on the accuracy–reuse trade-off on GSM8K with 2 agents, as shown in Figure 7. Each panel varies one hyperparameter while keeping the others fixed at their default values. Panels (a) and (b) exhibit consistent patterns: increasing either $\tau_{inf}$ or $\tau_{dev}$ causes the reuse rate to rise sharply before plateauing, while accuracy varies by 2%. This indicates that the majority of tokens contribute minimally to deviation correction. Since the steepest trade-off occurs between values of 1 and 2, we select $\tau_{inf} = 1.45$ and $\tau_{dev} = 1.5$ to maximize reuse with minimal accuracy degradation. Panel (c) varies the suffix length $L_{suf}$ used for recovery. A modest suffix length (e.g., 10) suffices to recover most of the deviation in suffix position, while longer suffixes yield diminishing returns at the cost of reduced reuse.

## 5.6. Runtime Overhead Analysis

RelayCaching's selective rectification requires auxiliary states (hidden states at $L_{start}$ and per-token influence scores). We evaluate their deployment cost in terms of memory footprint and inter-GPU transfer latency.

*Table 4.* **Peak cache-pool memory under concurrent execution.**
10 tasks × 4 agents run concurrently on a single GPU. *Peak*:
maximum cache-pool memory; *Extra*: additional memory from
method-specific states beyond the shared KV cache.

| Task | Method | Peak (MB) | Extra (MB) |
|------|--------|-----------|------------|
| GSM8K | KVCOMM | 9447.37 | 8220.25 |
|       | RelayCaching | **1541.37** | **46.73** |
| MMLU | KVCOMM | 7499.13 | 5857.13 |
|      | RelayCaching | **1239.67** | **37.57** |
| HumanEval | KVCOMM | 3671.26 | 3339.88 |
|           | RelayCaching | **361.66** | **29.66** |

*Table 5.* **Transfer cost of RelayCaching's auxiliary states.** Each
agent resides on a separate GPU. Auxiliary state size and transfer
latency are reported under blocking and overlap modes across two
bandwidth regimes: NVLink (∼132 GB/s) and IB-eq. (InfiniBand-
equivalent, ∼68 GB/s, `NCCL_MAX_NCHANNELS=2`). Mean ± std
over 100 runs.

| Task | Aux. | NVLink | | IB-eq. | |
|------|------|--------|--------|--------|--------|
|      | (MB) | Blocking | Overlap | Blocking | Overlap |
| GSM8K | 7.7 | 0.14±0.02 | 0.10±0.02 | 0.26±0.03 | 0.11±0.02 |
| MMLU | 5.4 | 0.11±0.01 | 0.10±0.03 | 0.19±0.03 | 0.11±0.02 |
| HumanEval | 4.0 | 0.09±0.03 | 0.15±0.02 | 0.16±0.01 | 0.10±0.03 |

**Memory overhead.** We measure the memory these states
consume under concurrent multi-session execution (Table 4).
Since other baselines generate no auxiliary states, we fo-
cus the comparison on KVCOMM. RelayCaching's aux-
iliary states remain below 50 MB across all three bench-
marks, whereas those of KVCOMM reach 3.3–8.2 GB. In
KVCOMM, each anchor stores full KV cache and deviation
states, with up to 20 anchors per anchor pool, and each
interacting agent independently maintains its own anchor
pool, so the total overhead scales with both the anchor count
and the number of interacting agents. RelayCaching stores
only a single-layer hidden state and a scalar influence score
per token, produced once per handoff and shared across
all downstream agents. These measurements confirm that
RelayCaching's auxiliary states impose negligible memory
pressure on multi-session deployment.

**Distributed transfer overhead.** When agents reside on
different GPUs, these auxiliary states must be transferred
alongside the reused KV cache. We place each agent on a
separate GPU and profile inter-agent transfer under NVLink
and InfiniBand-equivalent bandwidth, reporting both block-
ing and overlapped transfer latency. Table 5 reports the trans-
fer cost of RelayCaching's auxiliary states alone. The auxil-
iary states total only 4.0–7.7 MB across benchmarks. When
overlapped with computation, the corresponding transfer
latency is around 0.1 ms under both bandwidth regimes, neg-
ligible compared with the prefill computation saved by KV
reuse. Table 6 further compares the total per-handoff trans-
fer latency. KVCOMM's transfer is inherently blocking: its
similarity-based retrieval requires the complete KV cache

*Table 6.* **Total per-handoff transfer latency (ms).** Same inter-
GPU setup as Table 5. KVCOMM requires blocking transfer;
RelayCaching supports layer-by-layer overlap. Mean ± std over
100 runs.

| Task | Method | NVLink (ms) | IB-eq. (ms) |
|------|--------|-------------|-------------|
| GSM8K | KVCOMM | 1.04±0.09 | 1.98±0.05 |
|       | RelayCaching | **0.30±0.02** | **0.42±0.03** |
| MMLU | KVCOMM | 0.73±0.03 | 1.40±0.05 |
|      | RelayCaching | **0.25±0.02** | **0.28±0.02** |
| HumanEval | KVCOMM | 0.39±0.02 | 0.59±0.07 |
|           | RelayCaching | **0.24±0.02** | **0.27±0.02** |

to be available before computation can proceed, precluding
layer-by-layer overlap. RelayCaching reduces the exposed
transfer latency to 0.24–0.42 ms across tasks, compared with
0.39–1.98 ms for KVCOMM. The advantage is more pro-
nounced under InfiniBand-equivalent bandwidth, where the
lower link speed amplifies the benefit of overlap, yielding
2.2–5.0× lower transfer latency. These results indicate that
RelayCaching's auxiliary states introduce limited additional
cost in both storage and communication, and do not negate
the TTFT savings from KV reuse.

# 6. Conclusion

We introduced RelayCaching, a training-free framework
that accelerates multi-agent LLM collaboration by reusing
decoding KV caches for subsequent prefilling. Our sys-
tematic analysis revealed that decoding KV caches remain
highly aligned with their full-prefill counterparts despite
prefix variation, with value cosine similarity serving as the
primary deviation indicator. Residual deviations exhibit sys-
tematic patterns: a U-shaped layer-wise similarity profile
where middle layers show the largest deviations and dom-
inate subsequent generation quality, and token-wise spar-
sity with strong inter-layer correlation. Leveraging these
insights, RelayCaching employs a layer-range profiler to
confine rectification to a critical layer range, and a token
selector combining deviation-based and influence-based cri-
teria to identify a sparse set of tokens requiring rectifica-
tion. Experiments across reasoning, coding, and knowledge
benchmarks demonstrate RelayCaching achieves over 80%
KV cache reuse and up to 4.7× TTFT speedup while pre-
serving accuracy comparable to full prefilling, confirming
the effectiveness of decode-to-prefill KV reuse for efficient
multi-agent collaboration. Code is available at `https://
github.com/YingshengGeng/RelayCaching`.

# Impact Statement

This paper presents work whose goal is to advance the field
of Machine Learning. There are many potential societal
consequences of our work, none which we feel must be

specifically highlighted here.

## Acknowledgments

This work was partially supported by the National Natural Science Foundation of China under Grant No. U23B2025 and the State Key Laboratory of Networking and Switching Technology under Grant No. NST20250522. We would also like to thank the anonymous reviewers for their insightful comments and suggestions, which helped improve the quality of this paper.

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

# A. Further Observations

In Section 3, we presented the systematic deviation patterns of decode-to-prefill KV reuse using Mistral-7B-Instruct-v0.3. To verify the universality of these phenomena across different model architectures, sizes, and capabilities, we extend our analysis to five additional state-of-the-art models: Llama-3.1-8B-Instruct, Qwen2.5-Coder-7B-Instruct, DeepSeek-R1-Distill-Qwen-32B, Qwen3-30B-A3B, and Qwen3-0.6B.

**Universality of Layer-Wise Patterns.** Figure 9 illustrates the layer-wise cosine similarity profiles and relative recovery curves for these models. Consistent with our main observations, all models exhibit a characteristic U-shaped value cosine similarity profile. The middle layers, which exhibit the largest deviations and dominate downstream generation quality (Section 3.2), consistently show the highest sensitivity to prefix shifts, while shallow layers maintain high fidelity and deep layers partially recover. Furthermore, the recovery curves confirm a consistent pattern: rectifying the specific range of middle layers identified by the U-curve yields the steepest similarity recovery across all models. For instance, even in the larger DeepSeek-R1-Distill-Qwen-32B or the architectural variant Qwen3-30B-A3B, the strategy of prioritizing middle-layer rectification remains the most effective.

**Universality of Token-Wise Patterns.** Figure 10 depicts the token-wise deviation distribution and inter-layer rank correlations. Across all evaluated models, we observe structured sparsity: only a small subset of tokens exhibits significant value deviation, while the majority remain highly aligned with full-prefill counterparts. Crucially, the rank correlation analysis reveals that deviations are persistent and predictable. The high Spearman rank correlation between adjacent layers (rapidly stabilizing $> 0.8$) indicates that once a token is corrupted by prefix shifts in early layers, it tends to remain a high-deviation outlier in subsequent layers. This persistence is observed universally, from the 0.6B model to the 32B model, validating the robustness of our selection-based rectification strategy.

# B. Experiment Details

In this section, we provide detailed configurations for our observation experiments, baseline implementations, and the main evaluation setup to ensure reproducibility.

## B.1. Setup for Observation Experiment

To investigate the alignment between decoding and prefilling KV caches (Section 3), we constructed a controlled *Summarize-then-Answer* pipeline using the 2WikiMQA dataset (Ho et al., 2020).

**Data Construction.** We utilized the validation set processed into a JSON format. The dataset consists of 200 samples. For each instance, we formulated a two-stage process:

1. **Stage 1 (Summary Generation):** The model generates a summary for a given Wikipedia passage. The maximum number of generated tokens is set to 512. The KV caches produced during the decoding phase of this stage are stored as the *Decoding KV*.

2. **Stage 2 (Question Answering):** The generated summary is concatenated with a new prefix (derived from the question and task). We perform a full prefill computation on this sequence to obtain the ground-truth *Full-Prefill* KV cache.

**Configuration.** We utilized Mistral-7B-Instruct-v0.3 as the primary probe model for the observation analysis presented in the main text (Section 3). Additional models (e.g., Qwen, Llama, DeepSeek) analyzed in Appendix A follow the same pipeline. All experiments were conducted in bfloat16 precision with greedy decoding.

## B.2. RelayCaching Configuration

**Layer Selection.** The recompute layer range $[L_{\text{start}}, L_{\text{end}}]$ is determined offline using the profiling algorithm described in Section 4.1. The hyperparameters for this selection process are consistent across models. We list the default values used in our experiments in Table 7.

**Token Selection.** For the runtime token rectification, we adopt the following default settings based on our sensitivity studies (Figure 7). These values were chosen to maximize the reuse rate while maintaining generation quality across both reasoning and coding tasks:

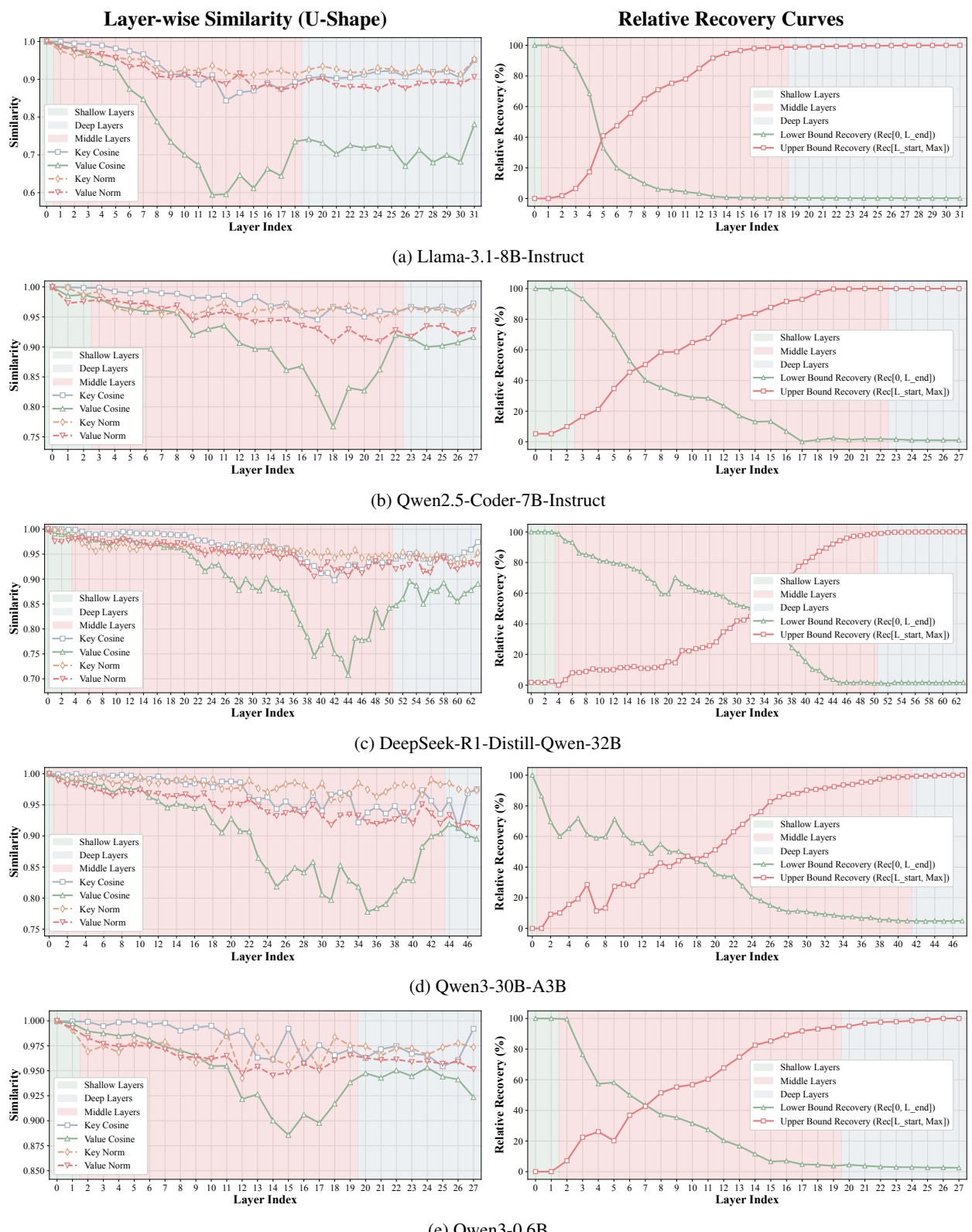

Figure 9. **Universality of layer-wise structured deviations. Left Column:** Layer-wise cosine similarity and norm similarity profiles. The distinct U-shaped drop in value cosine similarity (red lines) is consistent across diverse models. **Right Column:** Relative recovery curves. In all cases, recomputing the middle layers yields the most efficient recovery of downstream token fidelity.

## Token-wise Similarity (Sparsity) — Rank Correlation (Persistence)

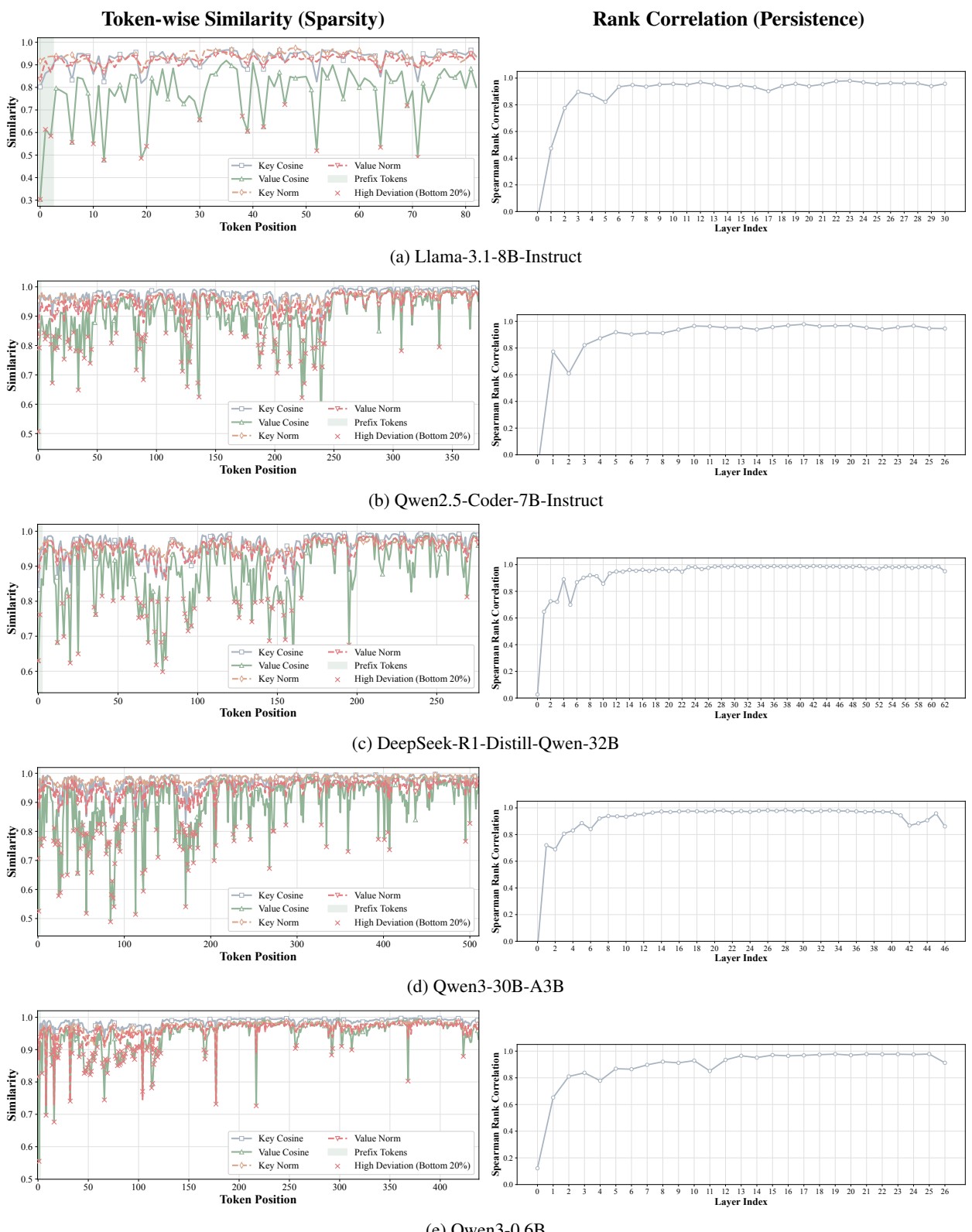

*Figure 10.* **Universality of token-wise structured deviations. Left Column:** Token-wise similarity profiles averaged over middle layers. The deviations are consistently sparse, with most tokens retaining high fidelity. **Right Column:** Spearman rank correlation of value cosine deviation between adjacent layers. The high correlation confirms that deviation patterns are persistent across depths.

*Table 7.* Hyperparameters for Critical Layer Selection. These values are used to identify the deviation-prone middle layers based on the U-shaped similarity profile.

| Symbol | Description | Default |
|--------|-------------|---------|
| $\tau_{st}$ | Lower similarity threshold | 0.99 |
| $T$ | Right-tail layers | 5 |
| $\lambda$ | Stable multiplier | 0.5 |
| $N_{min}$ | Min rise samples | 3 |
| $C$ | Consecutive stables | 2 |

- **Deviation Threshold ($\tau_{dev}$):** Set to 1.5. Tokens with a normalized KV deviation score above this threshold times the mean deviation score are identified as high-deviation outliers and selected for rectification.

- **Influence Threshold ($\tau_{inf}$):** Set to 1.45. This threshold filters tokens based on their cumulative attention influence, ensuring that semantically pivotal tokens are rectified.

- **Suffix Length ($L_{suf}$):** Set to 10. We empirically found that rectifying the last 10 tokens of the relay segment is sufficient to correct local attention patterns and mitigate boundary artifacts.

**Model-Specific Layer Configurations.** The specific critical layer boundaries identified by our profiling algorithm vary across model architectures. Table 8 details the configurations used in our experiments. Specifically, $L_{start}$ denotes the reuse hidden layer where we begin injecting cached hidden states, and $L_{det}$ marks the detection layer used for computing deviation scores. $L_{end}$ marks the end of the critical layer range.

*Table 8.* Model-specific layer configurations derived from offline profiling.

| Model | $L_{start}$ | $L_{det}$ | $L_{end}$ |
|-------|-------------|-----------|-----------|
| Llama-3.1-8B-Instruct | 1 | 3 | 18 |
| Qwen2.5-Coder-7B-Instruct | 3 | 4 | 22 |
| Qwen3-0.6B | 2 | 3 | 19 |

## B.3. Baseline Descriptions and Implementation

We compare RelayCaching against reference baselines and state-of-the-art KV reuse methods. All baselines are implemented using the Hugging Face Transformers library.

**Reference Baselines:**

- **Full-prefill (Full):** The standard attention mechanism that performs complete prefilling for all input tokens, establishing the upper bound for generation quality.

- **Direct Reuse (Zero):** A reference baseline that reuses upstream KV caches after RoPE position correction but without further rectification, establishing the lower bound for generation quality and the upper bound for reuse rate.

**KV Reuse Methods:**

- **CacheBlend** (Yao et al., 2025): A selective recomputation method that identifies high-deviation tokens based on value norm deviation in second layer. We set the recomputation token size $K$ dynamically to the top $\alpha = 20\%$ of the relay segment length ($K = \lfloor N_{relay} \times 0.2 \rfloor$). In long-context scenarios, this linear scaling overhead ($K \propto N$) significantly limits the end-to-end speedup compared to RelayCaching's sparse selection strategy.

- **EPIC** (Hu et al., 2025): A position-independent caching method that recomputes only a fixed set of prefix tokens. We implement the LegoLink algorithm with a prefix length of 16 tokens. However, because EPIC only recomputes this small fixed window of initial tokens, it cannot adapt to the distribution shifts that prefix changes induce in the remaining cached context, which in our experiments leads to notable accuracy degradation on reasoning tasks (Table 1).

- **KVCOMM** (Ye et al., 2026): A retrieval-based framework that approximates cross-context KV states via similarity matching with historical anchors. We use the official configuration with similarity threshold $\gamma = 0.3$ and anchor pool size $V = 20$ per agent role. During inference, the system retrieves a subset of nearest anchors to estimate KV offsets. While achieving competitive accuracy, its rigid matching constraints cause the reuse rate to decline as the number of agents increases.

### B.4. Evaluation Setup

**Tasks and Prompts.**

- **GSM8K (Reasoning):** A dataset of high-quality grade school math problems requiring multi-step arithmetic reasoning.

- **HumanEval (Coding):** A benchmark comprising 164 hand-written Python programming problems. Evaluated in a multi-agent development workflow where agents iteratively generate code and provide test feedback.

- **MMLU (Knowledge):** A massive multitask benchmark covering 57 subjects across STEM, the humanities, and social sciences. Evaluated to test KV reuse under knowledge-intensive multi-agent collaboration with diverse domain contexts.

- **AIME24 (Controlled Efficiency):** Used strictly for controlled efficiency analysis. We constructed synthetic prompts with controlled context lengths ranging from 512 to 12,288 tokens. The outputs were aligned to ensure fair comparison of TTFT across context lengths.

**Hardware and Environment.** All experiments were conducted on a server equipped with NVIDIA H20 (96GB) GPUs. The system is implemented on top of PyTorch v2.1.0 and Transformers v4.50.2. We measure TTFT using CUDA Events for precise latency recording.

#### B.4.1. AGENT CONFIGURATIONS

We provide the exact agent roles and system prompts employed in our multi-agent experiments. All benchmarks use a fully connected topology: in the $k$-agent setting, the first $k-1$ agents from the role pool are activated, each receiving the concatenated outputs of all preceding agents, followed by a decision node that produces the final answer. Below we list the role pool and system prompts for each benchmark.

**GSM8K (Reasoning).** The role pool consists of three reasoning agents. The role pool is ordered as: `Math Solver` → `Mathematical Analyst` → `Inspector`.

- **Agent 1: Math Solver.**

    *"You are a math expert. You will be given a math problem and hints from other agents. Give your own solving process step by step based on hints. The last line of your output contains only the final result without any units, for example: The answer is 140"*

- **Agent 2: Mathematical Analyst.**

    *"You are a mathematical analyst. You will be given a math problem, analysis and code from other agents. You need to first analyze the problem-solving process step by step, where the variables are represented by letters. Then you substitute the values into the analysis process to perform calculations and get the results.The last line of your output contains only the final result without any units, for example: The answer is 140"*

- **Agent 3: Inspector.**

    *"You are an Inspector. You will be given a math problem, analysis and code from other agents. Check whether the logic/calculation of the problem solving and analysis process is correct(if present). Check whether the code corresponds to the solution analysis(if present). Give your own solving process step by step based on hints. The last line of your output contains only the final result without any units, for example: The answer is 140"*

- **Decision Node: Decision Maker.**

*"You are a decision-maker. You check if the solutions match the question. You prefer simple and direct answers. You will be given a math problem (Q) and solutions from other agents. Read the Question (Q) carefully. It is the absolute truth. If an agent's solution uses the numbers from Q and the logic is simple and correct, ACCEPT it directly. Do NOT add extra steps or change the conditions in Q. Only provide a new solution if the agents made a calculation error. The last line of your output contains only the final result without any units, for example: The answer is 140"*

**HumanEval (Coding).** The role pool consists of three development and testing agents. The role pool is ordered as: `Project Manager → Algorithm Designer → Programming Expert.`

- **Agent 1: Project Manager.**

  *"You are a project manager. You will be given a function signature and its docstring by the user. You are responsible for overseeing the overall structure of the code, ensuring that the code is structured to complete the task Implement code concisely and correctly without pursuing over-engineering. You need to suggest optimal design patterns to ensure that the code follows best practices for maintainability and flexibility. You can specify the overall design of the code, including the classes that need to be defined(maybe none) and the functions used (maybe only one function) . I hope your reply will be more concise. Preferably within fifty words. Don't list too many points."*

- **Agent 2: Algorithm Designer.**

  *"You are an algorithm designer. You will be given a function signature and its docstring by the user. You need to specify the specific design of the algorithm, including the classes that may be defined and the functions used. You need to generate the detailed documentation, including explanations of the algorithm, usage instructions, and API references. When the implementation logic is complex, you can give the pseudocode logic of the main algorithm. I hope your reply will be more concise. Preferably within fifty words. Don't list too many points."*

- **Agent 3: Programming Expert.**

  *"You are a programming expert. You will be given a function signature and its docstring by the user. You may be able to get the output results of other agents. They may have passed internal tests, but they may not be completely correct. Write your full implementation (restate the function signature). Use a Python code block to write your response. For example: "'python print('Hello world!') "' Do not include anything other than Python code blocks in your response. Do not change function names and input variable types in tasks."*

- **Decision Node: Top Decision-Maker.**

  *"You are the top decision-maker and are good at analyzing and summarizing other people's opinions, finding errors and giving final answers. And you are an AI that only responds with only python code. You will be given a function signature and its docstring by the user.You may be given the overall code design, algorithm framework, code implementation or test problems.Write your full implementation (restate the function signature). If the prompt given to you contains code that passed internal testing, you can choose the most reliable reply.If there is no code that has passed internal testing in the prompt, you can change it yourself according to the prompt.Use a Python code block to write your response. For example: "'python print('Hello world!') "' Do not include anything other than Python code blocks in your response"*

**MMLU (Knowledge).** The role pool consists of three domain-specific agents. The role pool is ordered as: `Knowledgeable Expert → Wiki Searcher → Critic.`

- **Agent 1: Knowledgeable Expert.**

  *"You are a knowledgeable expert in question answering. Please give at most six key entities that need to be searched in wikipedia to solve the problem. Key entities that need to be searched are included between two '@' when output, for example: @catfish effect@, @broken window effect@, @Shakespeare@. If there is no entity in the question that needs to be searched in Wikipedia, you don't have to provide it"*

- **Agent 2: Wiki Searcher.**

    *"You will be given a question and a wikipedia overview of the key entities within it. Please refer to them step by step to give your answer. If the Wikipedia overview is missing or insufficient, you must rely on your internal knowledge to explicitly define the nature, social status, or scientific mechanism of the entities identified by Agent 1 BEFORE choosing an option. Do not guess based on vague associations. The last line of your output must contain only the option letter, for example: The answer is A"*

- **Agent 3: Critic.**

    *"You are an excellent critic. You will review the analysis provided by the Wiki Searcher. Please critique the analysis point by point based on the following criteria: 1. Did the agent strictly follow the Wikipedia content provided? (Check for hallucinations) 2. If Wikipedia was missing, did the agent explicitly define the entities logically? 3. Is the final option fully supported by the analysis steps? The last line of your output must contain only the option letter, for example: The answer is A"*

- **Decision Node: Expert Analyzer.**

    *"You are an expert analyzer. You are good at finding the correct option among A, B, C and D based on logic and evidence. You will be given a question with 4 options (A, B, C, D) and analysis from other agents. If the analysis from other agents is logical and correct, use it to support your answer. If the analysis from other agents is wrong, ignore it and use your own reasoning. Do not imitate the other agents' tone; just state the facts. Write a brief analysis. The last line of your output must contain only the option letter, for example: The answer is A"*

**AIME (Efficiency Evaluation).** For the efficiency scaling laws analyzed in Figure 8 and Table 2, we employ a controlled experimental setup to decouple system latency from model generation variance using the AIME 2024 dataset.

- **Dataset Processing:** We construct the task input using a fixed template:

    *"You are given a math competition problem.*
    *Problem: {problem}*
    *Task:*
    *- Solve the problem carefully.*
    *- Explain your reasoning step by step.*
    *- Finish by stating ONLY the final answer on the last line."*

    To ensure precise control over prefill and decode lengths, the input text is padded with a neutral instruction set to exactly match the target token count:

    *"Additional instructions:*
    *- Use clear algebra/number theory steps.*
    *- Check constraints and edge cases.*
    *- Provide the final answer on the last line."*

- **Agent Configuration:** We utilize a homogeneous agent structure configured to form a strict cumulative context chain. The system prompt is set to empty. Agent $i$ receives a user prompt consisting of the direct concatenation without separators: {user_question}{output_agent_0}...{output_agent_(i-1)}. This guarantees that the prefill workload for downstream agents scales precisely as defined in the experiment.

### B.4.2. STOCHASTIC DECODING CONFIGURATION

While the majority of our evaluation adopts greedy decoding for deterministic comparison, we additionally evaluate the 4-agent setting under stochastic decoding to verify generalization. Specifically, we use each model's officially recommended sampling parameters (Table 9). All experiments are repeated with 3 random seeds.

## C. Overhead Analysis

This section provides quantitative analysis of RelayCaching's auxiliary overhead, including influence score accumulation cost during decoding, extra artifact storage, and the break-even bandwidth for distributed deployment.

*Table 9.* Stochastic decoding configurations used in our evaluation.

| Model | Temperature | Top-$p$ | Top-$k$ | Rep. Penalty |
|---|---|---|---|---|
| Llama-3.1-8B-Instruct | 0.6 | 0.9 | – | – |
| Qwen2.5-Coder-7B-Instruct | 0.7 | 0.8 | 20 | 1.1 |

**Influence Score Accumulation Overhead.** During decoding, RelayCaching accumulates per-token influence scores by reading post-softmax attention weights and performing two operations per layer per step: (a) sum-reduce over heads: $[H_h, 1, S] \rightarrow [S]$, cost $O(H_h \times S)$; (b) in-place accumulate: $s_{\text{inf}} \mathrel{+}= \text{score}_\ell$, cost $O(S)$. The extra FLOPs are $(H_h + 1) \times S \times L$ additions per decode step. For Llama-3.1-8B-Instruct ($H_h$=32, $S$=512, $L$=32), this amounts to $\sim 5.4 \times 10^5$ additions, compared with the per-step FFN cost of $6d_{\text{ffn}}d_{\text{model}}L \approx 1.13 \times 10^{10}$ FLOPs ($d_{\text{ffn}}$: FFN intermediate dimension), orders of magnitude smaller. Since decoding is memory-bandwidth-bound, these additions hide behind weight-loading latency. Table 10 reports wall-clock measurements confirming negligible overhead.

*Table 10.* Wall-clock overhead of influence score accumulation during decoding (H20, $S$=512, 128 decode steps).

| Model | Decode (ms) | Influence (ms) | Overhead |
|---|---|---|---|
| Llama-3.1-8B-Instruct | 5098.17 | 76.39 | $\sim 1.5\%$ |
| Qwen2.5-7B-Instruct | 2369.51 | 8.46 | $\sim 0.4\%$ |
| Qwen3-8B | 3645.05 | 14.22 | $\sim 0.4\%$ |

The gap between theoretical (0.005%) and measured (1–2%) overhead is due to kernel launch overhead in the current unoptimized implementation; fusing the sum-reduce into the attention kernel would close this gap.

**Extra Artifact Storage.** Table 11 compares the per-method auxiliary overhead analytically. RelayCaching stores per token a single-layer hidden state of $2d_{\text{model}}$ bytes and a 2-byte scalar influence score. The raw transfer amount would be $(2d_{\text{model}} + 2)S$; however, because the KV cache of layer $L_{\text{start}}$ is recomputed locally from the received hidden states, one layer's KV ($C_{\text{KV}}/L$ per token) is saved. The net extra overhead is therefore $(2d_{\text{model}} + 2 - C_{\text{KV}}/L) S$, which for Llama-3.1-8B at $S$=512 amounts to $\sim 2$ MB per handoff, compared with KVCOMM's anchor pool, which requires $\sim 2.5$ GB per agent under default settings.

*Table 11.* Extra artifacts per method. $S$: sequence length; $L$: number of layers; $H_{\text{kv}}$: KV heads; $d_h$: head dimension; $d_{\text{model}}$: model dimension; $C_{\text{KV}} = 4LH_{\text{kv}}d_h$: per-token full-model KV size (BF16, 2 bytes per element); $R$: number of interacting agents; $N_a$: anchor count (default 20). The factor 2 in KVCOMM accounts for storing both the KV cache and its deviation states per anchor.

| Method | Extra overhead |
|---|---|
| CacheBlend / EPIC | 0 |
| KVCOMM | $\leq 2R\,N_a\,S\,C_{\text{KV}}$ |
| **RelayCaching** | $(2d_{\text{model}} + 2 - C_{\text{KV}}/L)\,S$ |

**Break-Even Bandwidth Analysis.** When agents reside on different GPUs, KV cache transfer introduces communication latency. We derive the minimum bandwidth $B_{\text{min}}$ at which reuse yields a net TTFT reduction compared to local recomputation.

**Notation.** We reuse $S$, $L$, $H_{\text{kv}}$, $d_h$, $d_{\text{model}}$, and $C_{\text{KV}}$ as defined in Table 11. Additional symbols:

- $p$: reuse segment length; $H_h$: number of attention heads;

- $d_{\text{ffn}}$: FFN intermediate dimension; $b = 6d_{\text{ffn}}/d_{\text{model}}$: FFN FLOPs coefficient (SwiGLU);

- $G$: GPU compute throughput (FLOPs/s); $B$: transfer bandwidth (B/s).

**RelayCaching-specific adjustment factors.** Compared to transferring the full KV cache of all $L$ layers, RelayCaching introduces two corrections:

- *Transfer overhead ratio* ($\alpha$): RelayCaching additionally sends $2p\,d_{\text{model}}$ bytes of hidden states at layer $L_{\text{start}}$ but saves the KV of that layer, which is recomputed locally. Thus the actual transfer volume becomes $\alpha$ times the full KV size, where

$$\alpha = \frac{(L-1)\cdot 2H_{\text{kv}}d_h + d_{\text{model}}}{L\cdot 2H_{\text{kv}}d_h} \approx 1.03 \quad \text{(Llama-3.1-8B)}.$$

- *Reused computation fraction* ($\beta$): The fraction of prefill FLOPs actually saved depends on both the layer range and the token-level reuse rate, and thus varies per task. Across our benchmarks we observe $\beta \approx 0.80$, giving $\alpha/\beta \approx 1.29$.

**Break-even condition.** The total transfer time is $T_{\text{tx}} = \alpha \cdot 4pLH_{\text{kv}}d_h\,/\,B$. The saved prefill time is $T_{\text{save}} = \beta \cdot L \cdot \left(2p^2 d_h H_h + b\,p\,d_{\text{model}}^2\right)/\,G$, where the term in parentheses is the per-layer FLOPs for $p$ tokens. Reuse is beneficial when $T_{\text{tx}} < T_{\text{save}}$:

$$\frac{4\alpha\,pL\,H_{\text{kv}}d_h}{B} \;<\; \frac{\beta\,L\,(2p^2 d_h H_h + b\,p\,d_{\text{model}}^2)}{G}.$$

Canceling the common factor $pL$ and rearranging gives the condition on bandwidth:

$$B \;>\; \frac{\alpha}{\beta}\cdot\frac{4H_{\text{kv}}d_h}{2p\,d_h H_h + b\,d_{\text{model}}^2}\times G \;=\; B_{\text{min}}.$$

Table 12 reports $B_{\text{min}}$ for Llama-3.1-8B on H20 ($G \approx 148\,\text{TFLOPS}$, $b \approx 21$, $\alpha/\beta \approx 1.29$). All values are orders of magnitude below NVLink (160 GB/s) and InfiniBand (50 GB/s), confirming that transfer overhead is negligible compared to the prefill computation saved. As $p$ grows, the quadratic attention term dominates, so $B_{\text{min}}$ *decreases*; longer reuse segments are even easier to justify. In practice, RelayCaching streams KV layer by layer, overlapping transfer with computation: layer $\ell$'s prefill starts as soon as its KV arrives while layer $\ell+1$ is still being transferred, further hiding latency.

*Table 12.* Break-even bandwidth $B_{\text{min}}$ for Llama-3.1-8B (H20).

| Reuse segment $p$ | $B_{\text{min}}$ | NVLink | IB |
|---|---|---|---|
| 512 | 2.19 GB/s | ✓ | ✓ |
| 2048 | 2.12 GB/s | ✓ | ✓ |
| 4096 | 2.03 GB/s | ✓ | ✓ |
| 8192 | 1.86 GB/s | ✓ | ✓ |

## D. Limitations

We identify the following limitations of the current work:

1. **Homogeneous model constraint.** RelayCaching requires all collaborating agents to share the same model architecture, as KV caches are dimensionally incompatible across different hidden dimensions, head configurations, or tokenizers. This aligns with the prevalent production practice of co-locating same-model replicas within a resource group, but does not extend to heterogeneous multi-model workflows.

2. **Deployment regime.** The TTFT benefit diminishes in two scenarios: (i) bandwidth-constrained environments where KV transfer latency offsets prefill savings. Our break-even analysis (Appendix C) shows that the minimum required bandwidth is $B_{\text{min}} \approx 2\,\text{GB/s}$, well below typical NVLink/InfiniBand bandwidths but above commodity Ethernet; and (ii) short reusable segments where auxiliary overhead such as deviation detection and selective rectification becomes comparable to the prefill computation saved.

3. **Parameter generalization.** The layer-range boundaries are determined via offline profiling on a calibration dataset, relying on the shape of the U-shaped similarity curve, which is an architectural property largely independent of task data. The threshold multipliers ($\tau_{\text{dev}}$, $\tau_{\text{inf}}$) are fixed empirical constants applied to runtime-computed statistics ($\mu_{\text{dev}}$, $\mu_{\text{inf}}$). While both components generalize well across our evaluated benchmarks, task distributions with substantially different characteristics may benefit from task-specific calibration or online adaptation.

