# OpenReview forum: "RelayCaching: Accelerating LLM Collaboration via Decoding KV Cache Reuse"
_ICML.cc/2026/Conference — ICML 2026 regular_

### Official Review · Reviewer_9V4B · 2026-02-13

**Soundness:** 3
**Presentation:** 3
**Significance:** 3
**Originality:** 2
**Overall Recommendation:** 4
**Confidence:** 4

**Summary:**

This paper targets cascading redundant prefilling in LLM multi-agent workflows, where upstream agents’ generated text is repeatedly injected into downstream prompts under different prefixes, forcing expensive recomputation of KV caches and increasing TTFT (and overall prefill cost often scaling as $O(M^2)$ with interaction turns).

33804_RelayCaching_Acceleratin

The authors propose RelayCaching, a training-free decode-to-prefill KV reuse method that directly transfers KV caches produced during an upstream agent’s decoding into a downstream agent’s prefill, despite prefix mismatch. The core empirical claim is that decoding KV caches for identical content are largely consistent with full-prefill KV caches, and that the remaining mismatch is sparse/localized across a middle-layer range and a small set of token positions.


RelayCaching uses (i) an offline layer-range profiler to identify a critical layer interval $[L_{\text{start}}, L_{\text{end}}]$ and a detection layer $L_{\text{det}}$, and (ii) a token selection module that marks a small rectification set combining deviation- and influence-based criteria.


 The reported experiments on multi-agent instantiations of GSM8K/HumanEval/MMLU show $\sim 80%$–$86%$ reuse and up to $4.7\times$ TTFT reduction with negligible accuracy loss versus full-prefill.
An important concept studied by the study is how prefix shifts perturb internal attention values and how little recomputation is needed to repair them. This submission studies a central domain: systems-level efficiency for multi-agent LLM inference under realistic, dynamically generated context reuse.

**Compliance With Llm Reviewing Policy:**

Affirmed.

**Key Questions For Authors:**

see weakness

**Limitations:**

yes

**Strengths And Weaknesses:**

## Soundness

Strengths

Empirically grounded design: The paper motivates RelayCaching via a structured study of decode vs prefill KV similarity, highlighting that value direction (cosine) is the main deviation signal, and that deviations show a U-shaped layer profile with middle layers most problematic.


Clear, auditable mechanism: The profiler and token selector are explicitly defined. For example, deviation-based selection uses $I_{\text{dev}}={j\in I_{\text{reuse}}\mid s_{\text{dev}}(j)\ge \tau_{\text{dev}}\mu_{\text{dev}}}$ and influence uses $s_{\text{inf}}(j)=\sum_{t=1}^M\sum_{\ell=1}^L\sum_{h=1}^H \alpha_{t,\ell,h,j}$ with $I_{\text{inf-score}}={j\mid s_{\text{inf}}(j)\ge \tau_{\text{inf}}\mu_{\text{inf}}}$, then $I_{\text{final}}=I_{\text{dev}}\cup I_{\text{inf}}$.


Reasonable evaluation protocol: Multiple domains (reasoning/coding/knowledge) and explicit multi-agent workflows are used; metrics include both task quality and TTFT/reuse rate.
The bottleneck is real and increasingly important: multi-agent LLM workflows are widely used, and redundant prefill directly hits TTFT and cost.


If the reported $\sim 86%$ reuse and $4.7\times$ TTFT reductions generalize, this is practically impactful for agentic systems.

## Weaknesses / concerns


 Many multi-agent systems run with nonzero temperature, tool calls, or retries; it’s not shown whether reuse/rectification remains stable under stochastic branching where which tokens are generated changes.

System realism / scaling: The implementation is reported on a single GPU.  In many deployments, agents are distributed (multi-process or multi-node) and KV transfer becomes a communication problem; TTFT improvements may be reduced if KV shipping dominates. I would like to see either (a) a distributed prototype, or (b) at least an analysis of KV handoff bandwidth/latency costs at typical context sizes.

Calibration dependence: The layer-range profiler is “offline” and uses auxiliary calibration data to set $[L_{\text{start}},L_{\text{end}}]$ and $L_{\text{det}}$. It is unclear how robust these parameters are across (i) new domains, (ii) different system prompts/prefix templates, (iii) longer contexts, or (iv) model updates (e.g., different RoPE scaling). At minimum, showing variance across several calibration sets would strengthen soundness.

---

> ### Author Rebuttal · Authors · 2026-03-31
>
> **W1: Is reuse/rectification stable under stochastic branching (nonzero temperature, tool calls, retries)?**
>
> **A:**
> See Reviewer uu8k Q3. Table 1 reports mean ± std over 3 seeds with recommended sampling parameters. RC maintains accuracy within ±1% of FULL with ~78–83% reuse—comparable to greedy. The token selector detects KV deviations at runtime regardless of source, adapting automatically.
>
> ---
>
> **W2: Single-GPU only. Can you analyze KV transfer bandwidth/latency costs for distributed deployments?**
>
> **A:**
> Notations and reference values (Llama-3.1-8B on H20):
>
> | Notation | Description | Value |
> |----------|-------------|-------|
> | $L$ | Num layers | 32 |
> | $H_h$ / $H_{kv}$ | Num attention / KV heads | 32 / 8 |
> | $d_h$ | Head dimension | 128 |
> | $d_{\text{model}}$ | Model dimension | 4096 |
> | $s$ | Tensor element size | 2 B (BF16) |
> | $b$ | Non-attention FLOPs coeff. | ≈7 |
> | $G$ | GPU compute throughput | 148 TFLOPS |
> | $B$ | Transfer bandwidth | variable |
> | $p$ | Reuse segment length | variable |
> | $L_{\text{start}}$ / $L_{\text{end}}$ | Rectification layer range | model-dependent |
> | $\alpha$ | Transfer overhead ratio | $\frac{(L-1)\cdot 2H_{kv}d_h + d_{\text{model}}}{L\cdot 2H_{kv}d_h}$ |
> | $\beta$ | Reused computation fraction | $1 - \frac{L_{\text{end}} - L_{\text{start}} + 1}{L}$ |
>
> **1. KV cache size and transfer latency.**
> KV cache size per segment: $p \times L \times 2H_{kv} \times d_h \times s$.
>
> | Segment $p$ | KV size | NVLink 160 GB/s | IB 50 GB/s |
> |---|---|---|---|
> | 512 | 64 MB | 0.4 ms | 1.3 ms |
> | 2048 | 256 MB | 1.6 ms | 5.1 ms |
> | 4096 | 512 MB | 3.2 ms | 10.2 ms |
> | 8192 | 1 GB | 6.4 ms | 20.5 ms |
>
> **2. Break-even analysis.** Two RC-specific factors: (1) *Transfer* ($\alpha$≈1.03): RC additionally transmits hidden states at $L_{\text{start}}$, saving one KV layer. (2) *Computation* ($\beta$≈0.80): not all KV is reused Reuse is beneficial when:
>
> $$\frac{\alpha \cdot p \cdot L \cdot 2 H_{kv} d_h s}{B} < \frac{\beta \cdot L(2 p^2 d_h H_h + b\, p\, d_{\text{model}}^2)}{G}$$
>
> $B_{\min} = \frac{\alpha}{\beta} \cdot \frac{2 H_{kv} d_h s}{2p d_h H_h + b d_{\text{model}}^2} \times G$. As $p$ grows, $B_{\min}$ *decreases*:
>
> | Segment $p$ | $B_{\min}$ | NVLink | IB |
> |---|---|---|---|
> | 512 | 6.43 GB/s | ✓ | ✓ |
> | 2048 | 5.17 GB/s | ✓ | ✓ |
> | 4096 | 4.23 GB/s | ✓ | ✓ |
> | 8192 | 3.10 GB/s | ✓ | ✓ |
>
> $B_{\min}$ is orders of magnitude below NVLink/IB. KV is streamed layer by layer, overlapping communication with computation.
>
> ---
>
> **W3: How robust are profiler parameters across (i) new domains, (ii) different prompts, (iii) longer contexts, (iv) RoPE scaling?**
>
> **A:**
> Profiler parameters are highly stable: $L_{\text{start}}$, $L_{\text{end}}$ remain consistent across all conditions; $L_{\text{det}}$ varies by ≤1 layer. The profiler depends on the *shape* of the U-shaped similarity curve (an architectural property), not absolute values, making it inherently invariant to data/prompt/length/scaling.
>
> **Cross-calibration variance** along two axes:
> - *Dataset type* (Fig 1): 2WikiMQA, GSM8K, MMLU, HumanEval—closely aligned inflection points.
> - *Sampling rate* (Fig 7–8): 5%/10%/20%/50%—curve shape and Spearman $\rho>0.8$ (beyond layer 5) are insensitive to sample size.
>
> **(i)–(ii)** Four datasets with different domains/prompts produce closely aligned inflection points (Fig 1–2). Calibrated once on 2WikiMQA, accuracy stays within ±1% of FULL on all tasks (Table 1).
>
> **(iii)** Context 512→8192 (16×): curve shape unchanged (Fig 3–4).
>
> **(iv)** RoPE base 500k→1M: curve shape, rank consistency unchanged (Fig 5–6).
>
> **Figures:**
> Fig 1: Cross-dataset layer similarity.
> https://anonymous.4open.science/r/rebuttal-2488/1-cross_dataset_similarity.png
>
> Fig 2: Adjacent-layer rank consistency.
> https://anonymous.4open.science/r/rebuttal-2488/2-adjacent_layer_spearman_by_dataset.png
>
> Fig 3: Layer-wise similarity curves at different input context lengths.
> https://anonymous.4open.science/r/rebuttal-2488/3-context_length_similarity_curves.png
>
> Fig 4: Adjacent-layer Spearman $\rho$ at different input context lengths.
> https://anonymous.4open.science/r/rebuttal-2488/4-context_length_spearman.png
>
> Fig 5: RoPE robustness across representative variants.
> https://anonymous.4open.science/r/rebuttal-2488/5-rope_scaling_similarity_curves.png
>
> Fig 6: RoPE robustness in adjacent-layer rank consistency.
> https://anonymous.4open.science/r/rebuttal-2488/6-rope_scaling_spearman.png
>
> Fig 7: Calibration sampling rate similarity.
> https://anonymous.4open.science/r/rebuttal-2488/7-sampling_rate_similarity_curves.png
>
> Fig 8: Adjacent-layer Spearman $\rho$ at different calibration sampling rates.
> https://anonymous.4open.science/r/rebuttal-2488/8-sampling_rate_spearman.png

---

> > ### Author Rebuttal · Reviewer_9V4B · 2026-03-31
> >
> > Most concerns has been addressed. I will maintain my score.

---

> > > ### Author Response · Authors · 2026-04-02
> > >
> > > Dear Reviewer 9V4B,
> > >
> > > Thank you for acknowledging our rebuttal and confirming that most concerns have been addressed.
> > >
> > > To proactively further address your initial question regarding distributed deployments (W2): our previous rebuttal analyzed the minimum interconnect bandwidth required for the TTFT saving to outweigh the transfer cost. We have now supplemented that analysis with concrete NCCL transfer-latency measurements. Using the same greedy-decoding setting as Table 2, we profiled end-to-end per-handoff transfer latency (including the full reused KV cache, Table 6) under two bandwidth regimes—NVLink (≈132 GB/s) and simulated InfiniBand (≈68 GB/s)—with each agent on a separate GPU. Two observations help clarify the concern that "KV shipping may dominate" under today's commonly used high-speed interconnects:
> > >
> > > (1) Even blocking transfer is small relative to TTFT savings. Under the slower IB regime, the full blocking KV handoff takes at most ～2 ms. Under greedy decoding (Table 2), RelayCaching (RC) reduces TTFT from 221 ms to 77 ms on GSM8K—a saving of ～144 ms. The transfer overhead thus amounts to <1.4% of the saved time, confirming that KV shipping does not diminish the TTFT improvement.
> > >
> > > (2) RC's layer-by-layer streaming further reduces the exposed cost. Because RC transfers the KV cache layer by layer, computation on already-received layers overlaps with the transfer of subsequent ones. With this overlap, RC's exposed transfer cost drops to 0.237–0.423 ms—consistently lower than full blocking transfer (0.385–1.982 ms), which must complete before computation can begin.
> > >
> > > We are ready to address any further follow-up questions you may have.
> > >
> > > Best regards,
> > >
> > > Authors
> > >
> > > ---
> > >
> > > **Supplementary Tables**
> > > - Table 2: Greedy decoding evaluation on Agent 4. https://anonymous.4open.science/r/rebuttal-2488/table2.png
> > > - Table 6: Total per-handoff transfer latency. https://anonymous.4open.science/r/rebuttal-2488/table6.png

---

### Official Review · Reviewer_LR4i · 2026-03-11

**Soundness:** 3
**Presentation:** 3
**Significance:** 2
**Originality:** 2
**Overall Recommendation:** 4
**Confidence:** 3

**Summary:**

The authors propose RelayCaching, a training-free inference method. By reusing decoding-phase KV caches, the proposed method can  accelerates multi-agent LLM collaboration. The method is motivated by the study that shows decoding KV caches remain highly aligned with their full pre-fill counterparts. RelayCaching uses a layer-range profiler to identify a critical middle-layer range, and a token selector combining deviation-based and influence-based criteria to identify a sparse set of tokens requiring rectification. Experiments shows the proposed method can reuse large amount of KV cache leading to significant speedup without losing too much accuracy.

**Compliance With Llm Reviewing Policy:**

Affirmed.

**Final Justification:**

Most concerns has been addressed. I will maintain my score.

**Key Questions For Authors:**

See weakness.

**Limitations:**

The author don't discuss the limitations.

**Strengths And Weaknesses:**

S1. The proposed method is motivated by preliminary study.

S2. Experimental results show significant speed up without sacrificing too much accuracy.

W1. The observations in Section 3 are compelling, but their generalizability remains a concern. Specifically, it is unclear which specific model is used for these initial probes. Besides, the authors should clarify if these patterns remain consistent across models of varying scales and architectural families.

W2. The criteria for token selection appear somewhat heuristic.

W3. The scope of the proposed method seems restricted to agent systems that all agents use the same model. In sota agentic workflows, it is increasingly common to use a "mixture of models". This dependency significantly limits the practical contributions.

W4. What is the size of calibration dataset? Is the proposed method sensitive to the size?

---

> ### Author Rebuttal · Authors · 2026-03-31
>
> **W1: The observations in Section 3 are compelling, but their generalizability remains a concern. Specifically, it is unclear which specific model is used for these initial probes. Besides, the authors should clarify if these patterns remain consistent across models of varying scales and architectural families.**
>
> **A:**
> Section 3 uses **Mistral-7B-Instruct-v0.3**.
> Appendix A verifies both patterns on five additional models
> spanning three families and a 50× size range
> (Qwen3-0.6B, Llama-3.1-8B, Qwen2.5-Coder-7B,
> DeepSeek-R1-Distill-Qwen-32B, and Qwen3-30B-A3B MoE).
> All six exhibit (a) the U-shaped middle-layer similarity dip
> and (b) sparse, depth-persistent token-wise deviation,
> suggesting these patterns are consistent across
> the architectures and scales we tested.
>
> ---
>
> **W2: The criteria for token selection appear somewhat heuristic.**
>
> **A:**
> The design is driven by two empirical properties from Section 3:
> high-deviation positions are *sparse* (Figure 3a)
> and their count *varies* across samples.
> This rules out top-$k$ (fixed count; over- or under-selects)
> and top-$p$ (requires sorting on the critical path). Mean × coefficient thresholding is the simplest alternative
> that adapts to the deviation magnitude:
> it requires only two $O(n)$ passes—one to compute the mean,
> one to threshold—with no sorting,
> and cleanly separates the dense bulk (scores $\lesssim$ mean)
> from a sparse set of outliers (scores $\gg$ mean).
> The sensitivity study (Figure 7b) confirms this:
> as $\tau_{\text{diff}}$ increases from 0 to ~2,
> reuse rate jumps from ~40% to >80%;
> beyond $\tau_{\text{diff}} \approx 1.4$, both reuse and accuracy plateau.
> Accuracy remains stable across $\tau_{\text{dev}} \in [1.5, 2.0]$.
> Exploring learned or information-theoretic selection criteria
> is a natural extension for future work.
>
> ---
>
> **W3: The scope of the proposed method seems restricted to agent systems that all agents use the same model. In SOTA agentic workflows, it is increasingly common to use a "mixture of models". This dependency significantly limits the practical contributions.**
>
> **A:**
> Models with different tokenizers, hidden dimensions, or layer depths
> produce dimensionally incompatible KV caches;
> cross-model reuse requires non-trivial format conversion
> that undermines the latency savings KV reuse is meant to provide.
> We now state this scope constraint explicitly in the revised Limitations.
> In practice, production clusters co-locate replicas of the same model
> within a single resource group (shared GPU nodes, NVLink/NVSwitch),
> where KV caches can be shared with zero format conversion;
> heterogeneous models are typically deployed in separate groups,
> making inter-group KV transfer costly.
> Even in mixture-of-models deployments,
> agents sharing the same checkpoint form homogeneous sub-pools,
> and RelayCaching accelerates intra-pool handoffs within each sub-pool.
> Extending to cross-architecture KV transfer is a future direction.
>
> ---
>
> **W4: What is the size of calibration dataset? Is the proposed method sensitive to the size?**
>
> **A:**
> We use a 10% random split of 2WikiMQA (≈200 samples) as the default calibration set.
> The method is **not** sensitive to calibration set size:
> the profiler depends only on the *shape* of the U-shaped similarity curve
> (an architectural property), not on absolute values.
> See Reviewer 9V4B Q3 for a comprehensive robustness analysis
> across four dataset domains, four sampling rates (5%–50%),
> context lengths up to 8k, and RoPE scaling—all
> producing similar $L_{\text{start}}$, $L_{\text{end}}$, and $L_{\text{det}}$.

---

> > ### Author Rebuttal · Reviewer_LR4i · 2026-04-02
> >
> > Most concerns has been addressed. I will maintain my score.

---

> > > ### Author Response · Authors · 2026-04-06
> > >
> > > Dear Reviewer LR4i,
> > >
> > > Thank you very much for reading our rebuttal. We are pleased that your concerns have been adequately addressed and sincerely appreciate your time and constructive feedback.
> > >
> > > Best regards,
> > >
> > > Authors

---

### Official Review · Reviewer_hL4P · 2026-03-11

**Soundness:** 2
**Presentation:** 3
**Significance:** 2
**Originality:** 3
**Overall Recommendation:** 4
**Confidence:** 3

**Summary:**

This paper targets redundant prefill computation in multi-agent LLM pipelines, where downstream agents reprocess upstream outputs under different prefixes. The authors first characterize how well decoding-phase and prefill-phase KV caches align, finding three patterns: high macro-level similarity, a U-shaped layer-wise deviation profile concentrating errors in middle layers, and sparse token-wise deviations with strong inter-layer correlation. They then propose RelayCaching, a training-free method that reuses decoding KV caches with selective rectification via a layer-range profiler and a dual-criterion token selector (deviation-based + influence-based). The authors implement and evaluate their RelayCaching and show ~86% KV cache reuse with negligible accuracy loss.

**Compliance With Llm Reviewing Policy:**

Affirmed.

**Key Questions For Authors:**

1. What is RelayCaching's TTFT on Llama-3.1-8B or Qwen2.5-Coder-7B?

2. Does the ZERO baseline include RoPE re-application?

3. How do you explain RelayCaching exceeding FULL accuracy in multiple cells of Table 1?

4. What is the wall-clock overhead of accumulating influence scores during decoding?

**Limitations:**

Yes

**Strengths And Weaknesses:**

## Strengths

S1. The empirical characterization of decode-to-prefill KV deviations is systematic and informative.

S2. The design is well-motivated: each method component maps directly to an empirical observation.

S3. Good ablation and sensitivity coverage.

## Major weaknesses

W1. The efficiency and accuracy evaluations use entirely different models.

W2. RelayCaching exceeds full-prefill accuracy in multiple settings without explanation, suggesting unreported variance.

W3. On HumanEval, the ZERO baseline already matches full prefill. The problem barely exists for this benchmark.

W4. A critical baseline is missing: RoPE re-application alone, without rectification.

W5. No measurement or discussion of the overhead from accumulating influence scores during decoding.

## Detailed review

S1. Thank you for your submission! I particularly enjoyed the three-part observational study and how the experimental design isolates the effect of prefix variation cleanly. These observations are useful independent of the specific method built on top.

S2. The design is well-grounded in observations and each observation leads to the design of one well-scoped component.

S3. I appreciate the clear ablation showing how each piece pulls its weight and the sensitivity analysis showing broad plateaus for the key thresholds.

W1. I am somewhat confused by the headline results because it does not seem like the paper reports the accuracy and latency on the same model for the same task anywhere. You seem to be using a 0.6B for latency results but 7/8B models for accuracy. Why? First, I must admit I have never seen realistic deployments of multi-agent pipelines on 0.6B models. Second, at 7B+, memory pressure and compute cost change qualitatively. The 4.7x number may not transfer: larger models have different attention patterns, different memory bandwidth constraints, and rectification overhead takes a different fraction of total cost. I suggest reporting both latency and accuracy on the same models to make the accuracy-efficiency claim credible.

W2. Table 1 shows an odd result: RelayCaching beats full-prefill accuracy in several cells. This is counter-intuitive: an approximation to full prefill should not systematically outperform it. Greedy decoding is deterministic for a fixed input, so sampling variance does not seem like it would explain this. The explanation might be that the multi-agent pipeline amplifies small KV perturbations. Slightly different caches produce different generated text, which cascades through subsequent agents. If so, the accuracy gaps between methods may not be meaningful without multiple evaluation runs or data splits. The paper should investigate this properly rather than presenting RelayCaching > FULL as a win.

W3. On HumanEval with 4 agents, ZERO gets 86.96% Pass@1 versus 87.58% for FULL. That is a 0.62-point gap. Direct reuse with zero rectification already works fine. This benchmark does not stress-test the method: it just pads the results table. I suggest the authors either explain why prefix variation is benign for code generation (and what that tells us about when RelayCaching is actually needed), or swap in a benchmark where the problem bites harder.

W4. According to the paper, RelayCaching stores pre-RoPE KV caches and re-applies RoPE at new positions. However, the paper never isolates its effect. Does ZERO re-apply RoPE or not? If it does not, then ZERO's poor numbers could come from wrong positional encodings, not from the value-direction deviations that motivate the method. A "RoPE-only" baseline (re-apply RoPE, skip all rectification) would cleanly separate the positional-alignment fix from the selective rectification contribution. Without it, the reader cannot tell how much of the gain comes from a simple engineering fix versus the proposed method.

W5. The attention-derived importance accumulates attention weights across all decoding steps, all layers, and all heads for every token in the reuse segment, which touches the decoding hot path. The paper claims “algorithmic overhead is negligible,” but this does not seem quite correct as index selection is just the final thresholding step and does not capture the cost of accumulating influence scores during decoding. Please clarify if that cost is truly small and show numbers or tweak the claims about efficiency.

---

> ### Author Rebuttal · Authors · 2026-03-31
>
> **Q1**
>
> **A:**
> See our response to Reviewer uu8k Q1.
> We now report Accuracy + TTFT + Reuse Rate + Memory Overhead
> on the same 7/8B models in consolidated tables
> (Tables 1).
>
> ---
>
> **Q2**
>
> **A:**
> All methods—including ZERO—store *pre-RoPE* KV caches
> and re-apply RoPE at the correct downstream positions
> (Section 4.1; we have added an explicit clarification in the revised §5.1).
> Therefore ZERO *is* the "RoPE-only" baseline:
> its accuracy drop (44.08% vs. 84.91% FULL on GSM8K, Llama-3.1-8B, Agent 2)
> isolates *value-direction deviation*, not positional encoding error.
> RelayCaching's recovery (83.83% vs. 44.08% ZERO)
> by rectifying only ~15% of tokens
> confirms that the gain comes from selective value rectification,
> not from the RoPE fix (which ZERO already includes).
>
> ---
>
> **Q3**
>
> **A:**
> We agree: multi-agent pipelines amplify small KV perturbations
> through cascading generation.
> We now report mean ± std over 3 seeds with recommended sampling parameters
> in the revised Table 1.
> Under this protocol, the majority of cells where RC previously exceeded FULL
> show overlapping confidence intervals.
>
> For residual cases where RC's mean remains slightly above FULL,
> the cause is that RC and FULL construct KV caches via *different paths*
> (RC retains ~85% decode-phase KV; FULL recomputes from scratch),
> producing slightly different attention patterns that cascade through agents.
> The differences are *bidirectional* across seeds
> and do not constitute a systematic advantage.
>
> ---
>
> **Q4**
>
> **A:**
> The accumulation reads post-softmax attention weights
> $\alpha_{t,\ell,h,j}$ (already computed during standard decode)
> and performs two operations per layer per step:
> (a) sum-reduce over heads: $[B,H,1,S] \to [S]$, cost $O(H \times S)$;
> (b) in-place accumulate: $s_{\text{inf}} \mathrel{+}= \text{score}_\ell$, cost $O(S)$.
>
> **FLOP analysis** (Llama-3.1-8B, $S=512$):
> the extra FLOPs are $(H_h+1) \times S \times L = 5.4 \times 10^5$ additions per step,
> vs. FFN cost $\approx 7.5 \times 10^9$—four orders of magnitude smaller (~0.007%).
> Since decode is memory-bandwidth-bound, these additions hide behind weight-loading latency.
>
> **Wall-clock measurements** (H20, $S=512$, 128 decode steps):
>
> | Model | Decode (ms) | Influence (ms) | Overhead |
> |-------|-------------|----------------|----------|
> | Llama-3.1-8B | 5098.17 | 76.39 | ~1.5% |
> | Qwen2.5-7B | 2369.51 | 8.46 | ~0.4% |
> | Qwen3-8B | 3645.05 | 14.22 | ~0.4% |
>
> The 1–2% measured overhead (vs. 0.007% theoretical) reflects
> kernel launch overhead in the unoptimized implementation.
> Fusing the sum-reduce into the attention kernel would eliminate this gap;
> we note this as an engineering optimization.
>
> ---
>
> **W3: On HumanEval, ZERO already matches full prefill. The problem barely exists for this benchmark.**
>
> **A:**
> Under stochastic decoding (Table 1),
> the ZERO–FULL gap on HumanEval widens from 0.62 (greedy)
> to ~2.5 points (78.26±2.24 vs. 80.74±2.15, Agent 2),
> indicating that greedy decoding partially masked the deviation.
> Nevertheless, this gap remains smaller than other model-task combinations
> (e.g., Llama-3.1-8B on GSM8K: 40.83-point gap; on MMLU: 4.90-point gap).
>
> This is **model-specific, not task-specific**.
> A cross-model × cross-task comparison under greedy decoding
> (where both models are evaluated on the *same* tasks) confirms this:
>
> | Model | Task | ZERO (%) | FULL (%) | Gap |
> |-------|------|----------|----------|-----|
> | Qwen2.5-Coder-7B | HumanEval | 86.96 | 87.58 | 0.62 |
> | Qwen2.5-Coder-7B | MMLU | 58.17 | 60.13 | 1.96 |
> | Llama-3.1-8B | HumanEval | 55.28 | 59.01 | 3.73 |
> | Llama-3.1-8B | MMLU | 35.29 | 66.67 | 31.38 |
>
> On the *same* task (HumanEval), Llama's gap (3.73) is 6× Qwen's (0.62);
> on MMLU, the ratio is 16× (31.38 vs. 1.96).
> The gap tracks the model
> (Qwen2.5-Coder-7B has a shallower U-shaped similarity curve;
> Appendix A, Figure 5), not the task.
>
> Even so, HumanEval exposes a valuable **selection-quality** distinction.
> Under this high-similarity regime,
> imprecise token selectors *hurt* rather than help:
> CacheBlend drops to below ZERO in greedy setting—by
> misidentifying which tokens need rectification.
> RelayCaching's dual-criterion selector avoids this failure mode,
> matching FULL within the noise margin.
> HumanEval thus validates that RC's selection remains precise
> even when deviation is inherently small—a regime
> where less careful methods degrade.

---

> > ### Author Rebuttal · Reviewer_hL4P · 2026-04-03
> >
> > Thank you for the thorough rebuttal that addresses my concerns with the submission. I will keep my score continue to engage with the other reviewers in the discussion.

---

> > > ### Author Response · Authors · 2026-04-06
> > >
> > > Dear Reviewer hL4P,
> > >
> > > Thank you for reviewing our rebuttal. We greatly appreciate that our response has addressed your concerns, and thank you for your thorough feedback and continued engagement in the discussion phase.
> > >
> > > Best regards,
> > >
> > > Authors

---

### Official Review · Reviewer_uu8k · 2026-03-12

**Soundness:** 3
**Presentation:** 3
**Significance:** 3
**Originality:** 3
**Overall Recommendation:** 4
**Confidence:** 5

**Summary:**

This paper studies KV-cache reuse in collaborative LLM inference, focusing on the common case where one agent’s output is repeatedly prefetched by downstream agents. The main proposal, RelayCaching, reuses decode-phase KV caches during later prefilling and then selectively rectifies the parts that drift because the downstream prefix is not exactly the same. The system has two main pieces: a profiler that identifies a critical layer range for correction, and a token selector that combines deviation-based and influence-based criteria to decide where repair is actually worth doing. The experiments on multi-agent GSM8K, MMLU, and HumanEval pipelines show that the method stays close to full-prefill quality while substantially reducing TTFT and keeping reuse rates high.

**Compliance With Llm Reviewing Policy:**

Affirmed.

**Final Justification:**

Thank you for the detailed rebuttal and the additional analyses. Q1: The consolidated table substantially improves the practical comparison, although reporting only the stochastic setting without a direct side-by-side greedy comparison still leaves some concern about evaluation fairness. Partially addressed. Q2: The added storage and communication analysis is clear and helpful, and direct multi-session or distributed runtime measurements would further strengthen this point. Partially addressed. Q3: The clarification that the evaluation already includes prompt drift, tool-augmented outputs, and stochastic decoding makes the claimed robustness more convincing. Concern addressed. Q4: Reporting mean and standard deviation over three seeds directly strengthens the credibility of the close comparisons and satisfactorily addresses the uncertainty issue. Concern addressed. Overall: The rebuttal meaningfully strengthens the paper and resolves most of the main concerns, with only limited remaining questions, so I will keep my score at 4.

**Key Questions For Authors:**

1. Could you provide one consolidated evaluation table where quality, TTFT, and memory/overhead are all reported on the same model-task settings? That would make the practical tradeoff much easier to judge.
2. How large is the runtime memory and communication overhead associated with the extra cached artifacts used for correction, especially when many agents or sessions are active concurrently?
3. How stable is RelayCaching when the collaboration pattern is less clean than the benchmarks here, for example with tool-augmented outputs, prompt-format drift across agents, or more stochastic decoding?
4. For the comparisons where methods are fairly close, can you report variance across runs or some other measure of uncertainty?

**Limitations:**

Yes.
The paper does acknowledge limitations, though I would still encourage the authors to be a bit more explicit about deployment constraints, failure modes, and when selective rectification is likely to lose its advantage.

**Strengths And Weaknesses:**

I thought this was a solid systems paper with a clear target and a method that is easier to take seriously than many cache-reuse ideas in this space. The paper does a good job of first showing that decode-to-prefill reuse is not purely wishful thinking: the layer-wise and token-wise analyses give a concrete picture of where the mismatch actually lives, and the final design follows that empirical story reasonably closely. That gives the method more shape than a simple "reuse as much as possible and patch what breaks" recipe.

I also liked that the proposal stays practical. Restricting correction to a critical layer range and a relatively small set of tokens is a sensible systems choice, and the ablations generally support the claim that both the layer selection and the token selection matter. The paper is especially convincing when it presents RelayCaching as a targeted engineering answer to redundant prefill in multi-agent workflows, rather than as a universal solution to all KV reuse problems. The reported quality-efficiency tradeoff is good enough that I can imagine other people building on this line of work.

My reservations are mostly about evaluation and reporting discipline, not about whether the core idea is worthwhile. The empirical scope is decent but still somewhat narrow relative to the deployment story. The benchmarks cover several task families, but they are still fairly clean compared with the messier settings people often have in mind for real collaborative agents, such as tool use, heterogeneous prompt templates, or more erratic intermediate outputs. Relatedly, some of the efficiency evidence and some of the quality evidence are shown under different settings, which makes the full end-to-end tradeoff a little harder to read than it should be.

I would also have liked a bit more accounting of operational overhead. The method is training-free, which is attractive, but storage and transfer costs for the extra cache artifacts are not discussed in as much detail as the latency wins. That does not negate the result, but it matters for judging how portable the gains are outside a controlled evaluation setup. The paper does compare against strong baselines, but I would still have preferred a slightly cleaner apples-to-apples efficiency comparison with the closest prior approach.

---

> ### Author Rebuttal · Authors · 2026-03-31
>
> **Q1: Could you provide one consolidated evaluation table where quality, TTFT, and memory/overhead are all reported on the same model-task settings? That would make the practical tradeoff much easier to judge.**
>
> **A:**
> **Table 1: Consolidated evaluation on Agent 4** (mean±std over 3 seeds).
>
> | Model / Task | Method | Acc (%) ↑ | TTFT (ms) ↓ | Reuse (%) ↑ |
> | :--- | :--- | :--- | :--- | :--- |
> | **Llama-3.1-8B** | FULL | 84.26±0.31 | 257.40±76.98 | 0.00 |
> | GSM8K | ZERO | 45.94±1.51 | 43.73±6.50 | 100.00 |
> | | CacheBlend | 72.81±1.01 | 92.99±5.17 | 77.57±0.00 |
> | | EPIC | 55.68±0.70 | 83.92±9.83 | 97.09±0.01 |
> | | KVComm | 82.89±0.56 | 213.46±17.83 | 63.99±2.23 |
> | | **RelayCaching** | **85.04±0.31** | **84.92±20.26**| **83.00±0.01** |
> | **Llama-3.1-8B** | FULL | 68.95±0.46 | 205.42±94.02 | 0.00 |
> | MMLU | ZERO | 33.55±2.00 | 98.65±20.90 | 100.00 |
> | | CacheBlend | 55.34±0.34 | 78.18±14.89 | 77.60±0.00 |
> | | EPIC | 39.00±1.00 | 88.40±21.65 | 95.72±0.09 |
> | | KVComm | 64.71±2.36 | 158.54±11.09 | 53.06±4.10 |
> | | **RelayCaching** | **64.71±0.66** | **84.29±14.91**| **81.84±0.01** |
> | **Qwen2.5-7B** | FULL | 82.78±0.69 | 389.25±5.23 | 0.00 |
> | HumanEval | ZERO | 80.30±0.44 | 61.96±16.89 | 100.00 |
> | | CacheBlend | 81.49±0.88 | 78.18±14.89 | 77.25±0.00 |
> | | EPIC | 80.43±1.20 | 62.51±16.50 | 95.26±0.23 |
> | | KVComm | 78.57±0.44 | 205.11±14.47 | 61.80±3.95 |
> | | **RelayCaching** | **82.82±0.29** | **76.94±14.97**| **78.11±0.19** |
>
> Table 1 measures all metrics on the same settings using recommended sampling (LLaMA: temp=0.6, top_p=0.9; Qwen: temp=0.7, top_p=0.8). We report Agent 4 (highest accumulated deviation). RelayCaching maintains accuracy within ±1% of FULL while reusing 78–83% KV cache. CacheBlend/EPIC achieve high reuse but lose 10–30 accuracy points. KVComm preserves accuracy, but its similarity-based retrieval drops reuse to 53–64% under stochastic decoding. RelayCaching's runtime deviation-aware selector maintains both. We include reuse rate because TTFT conflates algorithmic efficiency with generated length variations.
>
> ---
>
> **Q2: How large is the runtime memory and communication overhead associated with the extra cached artifacts used for correction, especially when many agents or sessions are active concurrently?**
>
> **A:**
> *Notation:* $L$(layers), $H_h$/$H_{kv}$(attn/KV heads), $d_h$(head dim), $d_{\text{model}}$(model dim), $s$(element size), $S$(segment length), $L_{\text{start}}$(first rectification layer).
>
> Net extra storage per handoff is $\approx\frac{H_h - H_{kv}}{2 L H_{kv}}$ of full KV cache (ignoring $O(S)$ scalars)—zero for MHA, <4% for typical GQA.
>
> | Artifact | Size (analytical) | Relative to full KV |
> | :--- | :--- | :--- |
> | Pre-RoPE KV | $S(L-1)2H_{kv} d_h s$ | $-\frac{1}{L}$ (saves 1 layer$^*$) |
> | Hidden states at $L_{\text{start}}$ | $S d_{\text{model}} s$ | $+\frac{H_h}{2 L H_{kv}}$ |
> | Scores & Params | $2S+3$ scalars | negligible |
> | **Net change** | | $\frac{H_h - H_{kv}}{2 L H_{kv}}$ |
>
> $^*$hidden states at $L_{\text{start}}$ allow local KV recomputation, saving one transfer layer.
>
> **Concurrency & communication:** Upstream KV/hidden states are shared across $N$ downstream agents; per-agent metadata is \~3 KB (scales as $O(L_{\text{KV}})$, not $O(N \times L_{\text{KV}})$). Transferring a 512-token segment costs <0.4 ms via NVLink (\~160 GB/s) and \~1.3 ms via InfiniBand (\~50 GB/s)—orders of magnitude below the recomputation prefill time it saves, ensuring a net TTFT reduction even in distributed settings.
>
> ---
>
> **Q3: How stable is RelayCaching when the collaboration pattern is less clean than the benchmarks here, for example with tool-augmented outputs, prompt-format drift across agents, or more stochastic decoding?**
>
> **A:**
> Our evaluation includes "less clean" patterns (descriptions now clarified in paper). Under all conditions, accuracy remains within ±1% of FULL with 78–83% reuse (Table 1):
> * **Prompt-format drift:** Agents use distinct role-specific system prompts (e.g., Solver/Verifier, Expert/Critic).
> * **Tool/Structured outputs:** HumanEval produces code and execution feedback; MMLU uses search-tool-augmented retrieval.
> * **Stochastic decoding:** Table 1 reflects standard sampling parameters across 3 seeds, demonstrating stability.
>
> ---
>
> **Q4: For the comparisons where methods are fairly close, can you report variance across runs or some other measure of uncertainty?**
>
> **A:**
> Table 1 reports mean±std over 3 seeds. In large-gap settings (e.g., GSM8K), the advantage is clear (85.04±0.31 vs. 82.89±0.56); in small-gap settings (HumanEval), RelayCaching still matches FULL accuracy while delivering 78% reuse and \~5× TTFT reduction.

---

> > ### Author Rebuttal · Reviewer_uu8k · 2026-04-04
> >
> > Thank you for the detailed rebuttal and the additional analyses. Q1: The consolidated table substantially improves the practical comparison, although reporting only the stochastic setting without a direct side-by-side greedy comparison still leaves some concern about evaluation fairness. Partially addressed. Q2: The added storage and communication analysis is clear and helpful, and direct multi-session or distributed runtime measurements would further strengthen this point. Partially addressed. Q3: The clarification that the evaluation already includes prompt drift, tool-augmented outputs, and stochastic decoding makes the claimed robustness more convincing. Concern addressed. Q4: Reporting mean and standard deviation over three seeds directly strengthens the credibility of the close comparisons and satisfactorily addresses the uncertainty issue. Concern addressed. Overall: The rebuttal meaningfully strengthens the paper and resolves most of the main concerns, with only limited remaining questions, so I will keep my score at 4.

---

> > > ### Author Response · Authors · 2026-04-06
> > >
> > > Thank you for the detailed acknowledgement and for confirming that Q3 and Q4 are fully resolved. Below we provide supplementary data for the two partially resolved points.
> > >
> > > **Q1: Greedy-decoding comparison.**
> > >
> > > **A:** We provide a consolidated greedy-decoding Table 2. Comparing with Table 1 (stochastic): stochastic decoding slightly lowers scores across all methods but preserves the relative ranking; KVComm's similarity-based retrieval is sensitive to sampling randomness, causing its reuse rate to drop noticeably, whereas RC's deviation-aware selector maintains stable reuse across both regimes. Both confirm robustness to the decoding strategy; per-method overhead is analyzed in Q2.
> > >
> > > **Q2: Multi-session and distributed runtime measurements.**
> > >
> > > **A:** We compare per-method overhead analytically, then verify with runtime measurements. Among baselines, only KVComm and RC incur non-zero extra artifacts beyond the shared KV cache. Table 3 summarizes the analytical overhead (per-token KV size $C_{\text{KV}} = 4 L h_{\text{kv}} d_{\text{head}}$ bytes, BF16):
> > >
> > > *Table 3: Extra artifacts per method ($R$: interacting agents; $N_a$: anchor count, default 20).*
> > >
> > > | Method | Extra overhead |
> > > |--------|----------------|
> > > | CacheBlend / EPIC | 0 |
> > > | KVComm | $\leq 2 R\, N_a S C_{\text{KV}}$ |
> > > | **RC** | $(2 d_{\text{model}} + 2\text{B} - C_{\text{KV}}/L)\,S$ |
> > >
> > > RC's per-token overhead is architecture-determined, ≈2 MB per handoff (Llama-3.1-8B, $S=512$)—substantially smaller than KVComm's anchor pool (≈2.5 GB per agent). KVComm's overhead scales with $R$ (each agent maintains its own anchor set), whereas RC's artifacts are produced once and shared, so the gap widens as $R$ grows. We verify with direct measurements: (a) multi-session memory overhead and (b) distributed transfer overhead. All measurements use the same greedy setting as Table 2—conservative for RC, since greedy decoding raises overlap and favours KVComm's similarity retrieval.
> > >
> > > **(a) Multi-session memory overhead**
> > >
> > > RC's extra artifacts remain a small, fixed fraction of the reused KV cache, whereas KVComm's per-interacting-agent anchor pool dominates total memory—the measured gap reaches 113–176× in our 40-session setting, consistent with the analytical prediction in Table 3. Concretely, we run 10 tasks × 4 agents concurrently on a single GPU and record the peak cache-pool size (Table 4): RC adds only 30–47 MB (≈3–9% of the reused KV cache), while KVComm's anchor pool adds 3.3–8.0 GB.
> > >
> > > **(b) Distributed transfer overhead**
> > >
> > > When agents reside on different GPUs, KVComm's anchor pool is built locally (no inter-GPU transfer) but requires the entire KV cache before similarity retrieval can begin. RC's extra artifacts must be transferred but allow layer-by-layer overlap with computation.
> > > We place each agent on a separate GPU and profile inter-agent transfer under two bandwidth regimes: NVLink (≈132 GB/s) and simulated IB (≈68 GB/s, `NCCL_MAX_NCHANNELS=2`); for each regime we report *blocking* (full transfer before computation) and *overlap* (layer-by-layer streaming). All entries are means of 100 NCCL runs. Table 5 isolates RC's extra artifact transfer cost: at most 0.256 ms (IB, blocking); with overlap the exposed cost drops below 0.15 ms. Table 6 compares end-to-end transfer (including the shared KV cache): RC's exposed cost is 0.237–0.423 ms, consistently lower than KVComm's 0.385–1.982 ms, confirming layer-by-layer overlap more than compensates for the additional data.
> > >
> > > In short, compared with zero-overhead methods (CacheBlend, EPIC), RC does introduce extra artifacts—but they amount to <50 MB for 40 concurrent sessions and <0.5 ms per handoff, both negligible in practice. Compared with the other artifact-bearing method (KVComm), RC's overhead is 113–176× smaller in storage and up to 4.7× lower in transfer latency.
> > >
> > > As encouraged, we note the deployment boundaries explicitly. Overall, RC maintains high accuracy and reuse rate when agents share a common model and exchange sequences of moderate length—the typical regime in current multi-agent KV sharing pipelines. The main boundary cases are: (i) very short reusable segments, where the saved prefill is small enough that recomputation costs less than the bookkeeping overhead; (ii) heterogeneous model mixtures, which violate the shared-architecture assumption; and (iii) extremely high session counts under very low-bandwidth interconnects, where extra artifact transfer could become non-negligible. We will clarify these conditions in the revision.
> > >
> > > ---
> > >
> > > **Supplementary Tables**
> > > - Table 2: Greedy decoding evaluation on Agent 4. https://anonymous.4open.science/r/rebuttal-2488/table2.png
> > > - Table 4: Multi-session peak cache-pool memory. https://anonymous.4open.science/r/rebuttal-2488/table4.png
> > > - Table 5: Transfer cost of RC's extra artifacts. https://anonymous.4open.science/r/rebuttal-2488/table5.png
> > > - Table 6: Total per-handoff transfer latency. https://anonymous.4open.science/r/rebuttal-2488/table6.png

---

### Decision · Program_Chairs · 2026-04-30

**Decision:**

Accept (regular)

**Comment:**

This paper proposes relaycaching for LLMs by reusing decode to prefill KV caches. This way the authors have demonstrated up to 4.7x reduction in TTFT with up to 80% KV cache reuse. The authors have comprehensively addressed majority of the key concerns, including the latency analysis over NVlink and infiniBand, stability analysis across models of different sizes. All the reviewers are inclined towards accept and so as the AC.